# The Curse of Depth in Large Language Models

**Wenfang Sun**[1]* **Xinyuan Song**[2]* **Pengxiang Li**[3]* **Lu Yin**[4] **Yefeng Zheng**[1] **Shiwei Liu**[5]†

[1]Westlake University [2]Emory University [3]Dalian University of Technology
[4]University of Surrey [5]University of Oxford

## Abstract

In this paper, we introduce *the Curse of Depth*, a concept that highlights, explains, and addresses the recent observation in modern Large Language Models (LLMs) where nearly half of the layers are less effective than expected. We first confirm the wide existence of this phenomenon across the most popular families of LLMs such as Llama, Mistral, DeepSeek, and Qwen. Our analysis, theoretically and empirically, identifies that the underlying reason for the ineffectiveness of deep layers in LLMs is the widespread usage of Pre-Layer Normalization (Pre-LN). While Pre-LN stabilizes the training of Transformer LLMs, its output variance exponentially grows with the model depth, which undesirably causes the derivative of the deep Transformer blocks to be an identity matrix, and therefore barely contributes to the training. To resolve this training pitfall, we propose **LayerNorm Scaling (LNS)**, which scales the variance of output of the layer normalization inversely by the square root of its depth. This simple modification mitigates the output variance explosion of deeper Transformer layers, improving their contribution. Across a wide range of model sizes (130M to 7B), our experiments show that LNS consistently outperforms previous normalization and scaling techniques in enhancing LLM pre-training performance. Moreover, this improvement seamlessly carries over to supervised fine-tuning. All these gains can be attributed to the fact that LayerNorm Scaling enables deeper layers to contribute more effectively during training. Our code is available at LayerNorm-Scaling.

## 1 Introduction

Recent studies reveal that the deeper layers (Transformer blocks) in modern LLMs tend to be less effective than the earlier ones [54, 16, 32, 26]. On the one hand, this interesting observation provides an effective indicator for LLM compression. For instance, we can compress deeper layers significantly more [54, 29, 14] to achieve high compression ratios. Even more aggressively, entire deep layers can be pruned completely without compromising performance [33, 39].

On the other hand, having many layers ineffective is undesirable as modern LLMs are extremely resource-intensive to train, often requiring thousands of GPUs trained for multiple months, let alone the labor used for data curation and administration [1, 44]. Ideally, we want all layers in a model to be well-trained, with sufficient diversity in features from layer to layer, to maximize the utility of resources [26]. The existence of ill-trained layers suggests that there must be something off with current LLM paradigms. Addressing such limitations is a pressing need for the community to avoid the waste of valuable resources, as new versions of LLMs are usually trained with their previous computing paradigm which results in ineffective layers.

To seek the immediate attention of the community, we re-introduce the concept of *the Curse of Depth (CoD)* to systematically present the phenomenon of ineffective deep layers in various LLM families,

---

*Equal contribution.
†Corresponding author: `shiwei.liu@maths.ox.ac.uk`

39th Conference on Neural Information Processing Systems (NeurIPS 2025).

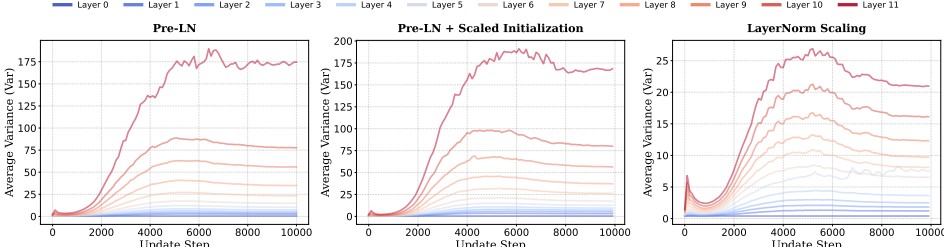

Figure 1: **Layerwise output variance.** This figure compares the output variance across various layers for different setups: (1) Pre-LN; (2) Pre-LN with Scaled Initialization [38, 35]; and (3) LayerNorm Scaling. The experiments are conducted on the LLaM-130M model trained for 10,000 steps. The proposed LayerNorm Scaling effectively controls the variance across layers.

to identify the underlying reason behind it, and to rectify it by proposing LayerNorm Scaling. We first state *the Curse of Depth* below.

**The Curse of Depth.** *The Curse of Depth* refers to the observed phenomenon where deeper layers in modern LLMs contribute significantly less (*but not nothing*) to learning and representation compared to earlier layers. These deeper layers often exhibit remarkable robustness to pruning and perturbations, implying they fail to perform meaningful transformations. This behavior prevents these layers from effectively contributing to training and representation learning, resulting in resource inefficiency.

**Empirical Evidence of CoD.** To demonstrate that CoD is a common phenomenon across prominent LLM families, we perform layer pruning experiments on Qwen3, LLaMA2, and DeepSeek. Specifically, we prune one layer at a time, without any fine-tuning, and directly evaluate the resulting pruned models on the MMLU benchmark [17], as shown in Figure 2. **Key findings:** (1) Most models, including the latest Qwen3, exhibit surprising resilience to the removal of deeper layers; (2) The number of layers that can be removed without causing significant performance drop increases with model size; (3) Representations in deeper layers are significantly more similar to each other than those in earlier layers.

**Identifying the Root Cause of CoD.** We theoretically and empirically identify the root cause of CoD as the use of Pre-Layer Normalization (Pre-LN) [3, 8], which normalizes layer inputs before applying the main computations, such as attention or feedforward operations, rather than after. Specifically, while stabilizing training, we observe that the output variance of Pre-LN accumulates significantly with layer depth (see Appendix F), causing the derivatives of deep Pre-LN layers to approach an identity matrix. This behavior prevents these layers from introducing meaningful transformations, leading to diminished representation learning.

**Mitigating CoD through LayerNorm Scaling.** We propose LayerNorm Scaling (LNS), which scales the output of Layer Normalization by the square root of the depth $\frac{1}{\sqrt{l}}$. LayerNorm Scaling effectively scales down the output variance across layers of Pre-LN. LNS consistently delivers better pre-training performance than Pre-LN with various model sizes from 130M to 7B. Unlike previous LayerNorm variants [26, 28], LayerNorm Scaling is simple to implement, requires no hyperparameter tuning, and introduces no additional parameters during training.[3] Furthermore, we show that the model pre-trained with LayerNorm Scaling achieves better performance on downstream tasks in self-supervised fine-tuning, all thanks to the more diverse feature representations learned in deep layers.

## 2 Empirical Evidence of the Curse of Depth

To empirically analyze the impact of layer normalization on the *Curse of Depth* in LLMs, we conduct a series of evaluations inspired by Li et al. [26], to compare Pre-LN and Post-LN models.

### 2.1 Experimental Setup

**Methodology:** We evaluate Pre-LN and Post-LN models by assessing the impact of layer pruning at different depths. Our hypothesis is that Pre-LN models exhibit diminishing effectiveness in deeper layers, whereas Post-LN models have less effective early layers.

---

[3]We found that combining LNS with Scaled Initialization diminishes the effectiveness of LNS. Therefore, we recommend removing the latter when applying LNS.

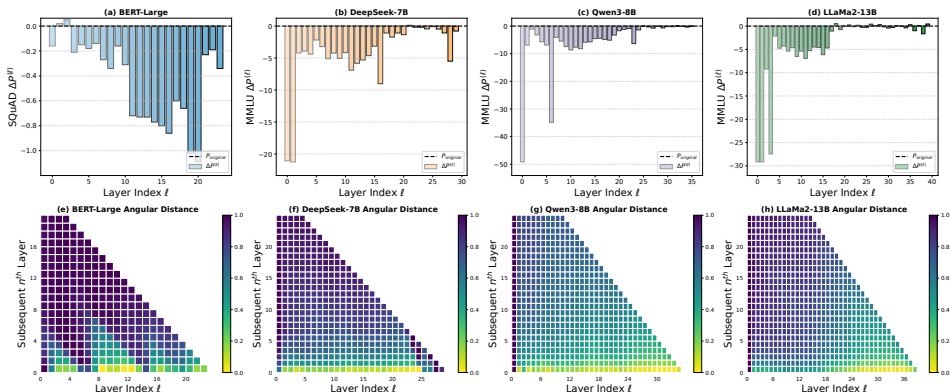

Figure 2: **Top:** Performance drop after removing a single layer without fine-tuning. **Bottom:** Angular distance from the initial layer $\ell$ (x-axis) and its subsequent $n^{\text{th}}$ layer (y-axis). The results demonstrate that in Pre-LN LLMs, deeper layers produce highly similar representations to their adjacent layers, and their removal results in minimal performance degradation. In contrast, Post-LN models show the opposite trend: deep layers contribute more substantially to model performance. *See Appendix D for controlled experiments, where we train custom LLMs from scratch to isolate Layer Normalization as the sole varying factor.*

**Models:** To verify this, we empirically quantify the contribution of individual layers to overall model performance across a diverse set of LLMs, including Qwen3 [42], LLaMA2 [44], DeepSeek [6], and BERT-Large [10]. These models were chosen to ensure architectural and application diversity. BERT-Large represents a Post-LN model, whereas the rest are Pre-LN-based. This selection enables a comprehensive evaluation of the effects of layer normalization across varying architectures and model scales.

**Evaluation Metric:** To empirically assess the impact of deeper layers in LLMs, we adopt two metrics, *Performance Drop* and *Angular Distance*, inspired by Gromov et al. [16], Li et al. [26].

*Performance Drop* $\Delta P^{(\ell)}$ quantifies the importance of each layer by measuring the performance change after its removal. A smaller $\Delta P^{(\ell)}$ indicates that the pruned layer contributes less to the model's overall performance. For BERT-Large, we evaluate using the SQuAD v1.1 dataset [36], while for other models, we use MMLU [17], a standard benchmark for multi-task language understanding.

*Angular Distance* $d(x^\ell, x^{\ell+n})$ quantifies the directional change between the input representations at layer $\ell$ and layer $\ell + n$ on a neutral pre-training dataset. Formally, given a token $T$, let $x_T^\ell$ and $x_T^{\ell+n}$ denote its input to layers $\ell$ and $\ell + n$, respectively. The angular distance is defined as:

$$d(x^\ell, x^{\ell+n}) = \frac{1}{\pi} \arccos\left( \frac{x_T^\ell \cdot x_T^{\ell+n}}{\|x_T^\ell\|_2 \|x_T^{\ell+n}\|_2} \right), \tag{1}$$

where $\| \cdot \|_2$ denotes the $L^2$-norm. To reduce variance, we report the average distance over 256K tokens sampled from the C4 dataset. Smaller values of $d(x^\ell, x^{\ell+n})$ indicate higher similarity between the two representations, suggesting limited transformation. Such layers can be considered redundant, as their removal minimally impacts the model's internal representations. Ideally, each layer should introduce meaningful representational shifts to fully leverage the model's capacity [53, 16].

**Experimental Results:** (1) Pruning deep layers in Pre-LN LLMs leads to negligible, and sometimes even positive, changes in performance, as shown in Figure 2-Top. Specifically, Figure 2 (b)–(d) reveals that a wide range of deeper layers—particularly beyond the 18th—can be pruned with minimal impact on performance. This indicates that deep layers in Pre-LN architectures contribute little to the model's overall effectiveness. In contrast, Figure 2 (a) shows that pruning deep layers in BERT-Large (a Post-LN model) leads to a substantial drop in accuracy, while pruning early layers has a relatively minor effect. (2) Pre-LN models exhibit decreasing angular distance in deeper layers, indicating highly similar representations, as shown in Figure 2-Bottom. For instance, the angular distance in DeepSeek-7B falls below 0.2 after the 18th layer. Qwen3-8B demonstrates a higher similarity, with nearly half of its layers exhibiting distances below 0.2 from their preceding layers. In LLaMA2-13B, the angular distance approaches zero across the final one-third of the network. These similar

representations align with the pruning results in Figures 2 (b)–(d), where pruning later layers has little effect, while pruning early layers significantly degrades performance.

## 3  Analysis of the Curse of Depth

**Preliminaries.** This paper primarily focuses on Pre-LN Transformer [3, 8]. Let $x_\ell \in \mathbb{R}^d$ be the input vector at the $\ell$-th layer of Transformer, where $d$ denotes the feature dimension of each layer. For simplicity, we assume all layers to have the same dimension $d$. The layer output $y$ is calculated as follows:

$$y = x_{\ell+1} = x'_\ell + \text{FFN}(\text{LN}(x'_\ell)), \tag{2}$$

$$x'_\ell = x_\ell + \text{Attn}(\text{LN}(x_\ell)), \tag{3}$$

where LN denotes the layer normalization function. In addition, the feed-forward network (FFN) and the multi-head self-attention (Attn) sub-layers are defined as follows:

$$
\begin{aligned}
\text{FFN}(x) &= W_2 \mathcal{F}(W_1 x), \\
\text{Attn}(x) &= W_O(\text{concat}(\text{head}_1(x), \dots, \text{head}_h(x))), \\
\text{head}_i(x) &= \text{softmax}\left( \frac{(W_{Qi}x)^\top (W_{Ki}X)}{\sqrt{d_{\text{head}}}} \right) (W_{Vi}X)^\top,
\end{aligned}
\tag{4}
$$

where $\mathcal{F}$ is an activation function, concat concatenates input vectors, softmax applies the softmax function, and $W_1 \in \mathbb{R}^{d_{\text{ffn}} \times d}$, $W_2 \in \mathbb{R}^{d \times d_{\text{ffn}}}$, $W_{Qi} \in \mathbb{R}^{d_{\text{head}} \times d}$, $W_{Ki} \in \mathbb{R}^{d_{\text{head}} \times d}$, $W_{Vi} \in \mathbb{R}^{d_{\text{head}} \times d}$, and $W_O \in \mathbb{R}^{d \times d}$ are parameter matrices, and $d_{\text{FFN}}$ and $d_{\text{head}}$ are the internal dimensions of FFN and multi-head self-attention sub-layers, respectively. $X \in \mathbb{R}^{d \times s}$, where $s$ is the input sequence length.

The derivatives of Pre-Ln Transformers are:

$$\frac{\partial \text{Pre-LN}(x)}{\partial x} = I + \frac{\partial f(\text{LN}(x))}{\partial \text{LN}(x)} \frac{\partial \text{LN}(x)}{\partial x}, \tag{5}$$

where $f$ here represents either the multi-head attention function or the FFN function. If the term $\frac{\partial f(\text{LN}(x))}{\partial \text{LN}(x)} \frac{\partial \text{LN}(x)}{\partial x}$ becomes too small, the Pre-LN layer $\frac{\partial \text{Pre-LN}(x)}{\partial x}$ behaves like an identity map. Our main objective is to prevent identity map behavior for very deep Transformer networks. The first step in this process is to compute the variance $\sigma_{x_\ell}^2$ of vector $x_\ell$.

### 3.1  Pre-LN Transformers

**Assumption 1.** *Let $x_\ell$ and $x'_\ell$ denote the input and intermediate vectors of the $\ell$-th layer. Moreover, let $W_\ell$ denote the model parameter matrix at the $\ell$-th layer. We assume that, for all layers, $x_\ell$, $x'_\ell$, and $W_\ell$ follow normal and independent distributions with mean $\mu = 0$.*

**Lemma 1.** *Let $\sigma_{x'_\ell}^2$ and $\sigma_{x_\ell}^2$ denote the variances of $x'_\ell$ and $x_\ell$, respectively. These two variances exhibit the same overall growth trend, which is:*

$$\sigma_{x_\ell}^2 = \sigma_{x_1}^2 \Theta\Big( \prod_{k=1}^{\ell-1} \Big( 1 + \frac{1}{\sigma_{x_k}} \Big) \Big), \tag{6}$$

*where the growth of $\sigma_{x_\ell}^2$ is sub-exponential, as shown by the following bounds:*

$$\Theta(L) \leq \sigma_{x_L}^2 \leq \Theta(\exp(L)). \tag{7}$$

Here, the notation $\Theta$ means: if $f(x) \in \Theta\big(g(x)\big)$, then there exist constants $C_1, C_2$ such that $C_1 |g(x)| \leq |f(x)| \leq C_2 |g(x)|$ as $x \to \infty$. The lower bound $\Theta(L) \leq \sigma_{x_\ell}^2$ indicates that $\sigma_{x_\ell}^2$ grows at least linearly, while the upper bound $\sigma_{x_\ell}^2 \leq \Theta(\exp(L))$ implies that its growth does not exceed an exponential function of $L$.

Based on Assumption 1 and the work of [41], we obtain the following:

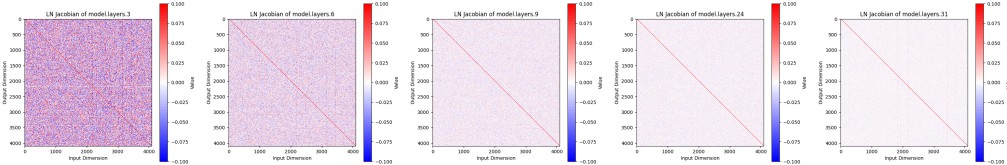

Figure 3: Visualization of the LayerNorm Jacobian matrices across different layers of a pre-trained LLaMA2-7B model. Each heatmap shows the token-averaged Jacobian at a specific layer. As depth increases, the Jacobians exhibit a pronounced diagonal dominance with vanishing off-diagonal entries, indicating that deep LayerNorm blocks increasingly approximate identity mappings.

**Theorem 1.** *For a Pre-LN Transformer with $L$ layers, using Equations* (2) *and* (3)*, the partial derivative $\frac{\partial y_L}{\partial x_1}$ can be written as:*

$$\frac{\partial y_L}{\partial x_1} = \prod_{\ell=1}^{L-1} \left( \frac{\partial y_\ell}{\partial x'_\ell} \cdot \frac{\partial x'_\ell}{\partial x_\ell} \right). \tag{8}$$

*The Euclidean norm of $\frac{\partial y_L}{\partial x_1}$ is given by:*

$$\left\| \frac{\partial y_L}{\partial x_1} \right\|_2 \leq \prod_{l=1}^{L-1} \left( 1 + \frac{1}{\sigma_{x_\ell}} A + \frac{1}{\sigma_{x_\ell}^2} B \right), \tag{9}$$

*where $A$ and $B$ are constants for the Transformer network. Then the upper bound for this norm is given as follows: when $\sigma_{x_\ell}^2$ grows exponentially, (i.e., at its upper bound), we have:*

$$\sigma_{x_\ell}^2 \sim \exp(\ell), \quad \left\| \frac{\partial y_L}{\partial x_1} \right\|_2 \leq M, \tag{10}$$

*where the gradient norm converges to a constant $M$. Conversely, when $\sigma_{x_\ell}^2$ grows linearly (i.e., at its lower bound), we have*

$$\sigma_{x_\ell}^2 \sim \ell, \quad \left\| \frac{\partial y_L}{\partial x_1} \right\|_2 \leq \Theta(L), \tag{11}$$

*which means that the gradient norm grows linearly in $L$.*

The detailed description of $A$ and $B$, as well as the complete proof, are provided in Appendix A.2. From Theorem 1, we observe that when the variance grows exponentially, as the number of layers $L \to \infty$, the norm $\left\| \frac{\partial y_L}{\partial x_1} \right\|_2$ is bounded above by a fixed constant $M$. This result implies that even an infinitely deep Transformer remains stable, and by the Weierstrass Theorem, the network is guaranteed to converge. Consequently, this implies that for very large $L$, deeper layers behave nearly as an **identity map** from $x_\ell$ to $y_\ell$, thereby limiting the model's expressivity and hindering its ability to learn meaningful transformations. This phenomenon is empirically illustrated in Figure 3, where we visualize the LayerNorm Jacobians of a pre-trained LLaMA2-7B model across depth, revealing a clear collapse toward identity mappings in deeper layers. This outcome is undesirable, therefore, we would instead prefer the variance to increase more gradually—e.g., linearly—so that $\left\| \frac{\partial y_L}{\partial x_1} \right\|_2$ exhibits linear growth. This observation highlights the necessity of appropriate variance control mechanisms, such as scaling strategies, to prevent excessive identity mappings and enhance network depth utilization.

## 4 LayerNorm Scaling (LNS)

To mitigate the abovementioned issue, we propose LayerNorm Scaling, a simple yet effective normalization strategy. The core idea of LayerNorm Scaling is to control the exponential growth of output variance in Pre-LN by scaling the normalized outputs according to layer depth. Specifically,

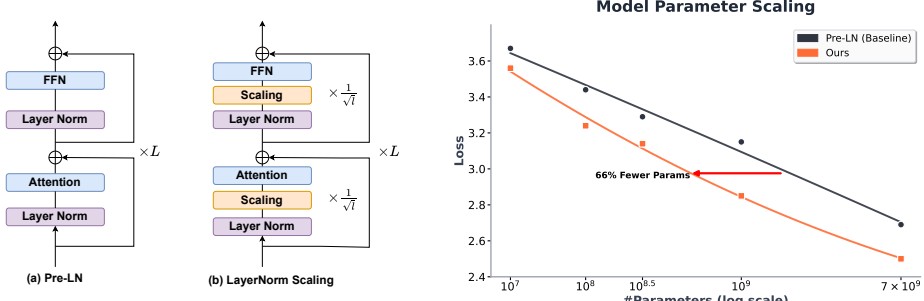

Figure 4: **Left:** Schematic diagrams of (a) Pre-LN and (b) LayerNorm Scaling. LayerNorm Scaling applies a scaling factor inversely proportional to the square root of the layer index $\ell$, preventing excessive variance growth. **Right**: Language modeling loss of scaling up parameter count up to 7B. All models are trained for 20B tokens using OLMo [15].

we apply a scaling factor inversely proportional to the square root of the layer index to scale down the output of LN layers, enhancing the contribution of deeper Transformer layers during training. LayerNorm Scaling is illustrated in the left part of Figure 4.

Formally, for a Transformer model with $L$ layers, the output of Layer Normalization in each layer $\ell$ is scaled by a factor of $\frac{1}{\sqrt{\ell}}$. Let $\mathbf{h}^{(\ell)}$ denote the input to Layer Normalization at layer $\ell$. The modified output is computed as:

$$\mathbf{h}^{(\ell)} = \text{LayerNorm}(\mathbf{h}^{(\ell)}) \times \frac{1}{\sqrt{\ell}}, \tag{12}$$

where $\ell \in \{1, 2, \ldots, L\}$. This scaling prevents excessive variance growth with depth, addressing a key limitation of Pre-LN. Unlike Mix-LN, which stabilizes gradients in deeper layers but suffers from training instability caused by Post-LN [34, 47], LayerNorm Scaling preserves the stability advantages of Pre-LN while enhancing the contribution of deeper layers to representation learning. Applying LayerNorm Scaling leads to a notable reduction of layerwise output variance as shown in Figure 1, resulting in a lower training loss. Moreover, compared with previous LayerNorm variants [26, 28], LayerNorm Scaling is hyperparameter-free, easy to implement, and does not introduce additional learnable parameters, making it computationally efficient and readily applicable to existing Transformer architectures. Our theoretical analysis in Appendix B shows that LNS slows down the variance growth from exponential to at most quadratic with depth, enabling more effective learning of deep layers.

## 5 Experiments

### 5.1 LLM Pre-training

To evaluate the effectiveness of LayerNorm Scaling, we follow the experimental setup of Li et al. [26], using the identical model configurations and training conditions to compare LNS with widely used normalization techniques, including Post-LN [34], DeepNorm [47], and Pre-LN [8]. In line with Lialin et al. [27] and Zhao et al. [58], we conduct experiments using LLaMA-based architectures with model sizes of 130M, 250M, 350M, and 1B parameters.

Table 1: Perplexity ($\downarrow$) comparison of various layer normalization methods.

| | LLaMA-130M | LLaMA-250M | LLaMA-350M | LLaMA-1B |
|---|---|---|---|---|
| Training Tokens | 2.2B | 3.9B | 6.0B | 8.9B |
| Post-LN [2] | 26.95 | 1409.79 | 1368.33 | 1390.75 |
| DeepNorm [47] | 27.17 | 22.77 | 1362.59 | 1409.08 |
| Mix-LN [26] | 26.07 | 21.39 | 1363.21 | 1414.78 |
| Pre-LN [3] | 26.73 | 21.92 | 19.58 | 17.02 |
| Pre-LN + LayerNorm Scaling | **25.76** | **20.35** | **18.20** | **15.71** |

The architecture incorporates RMSNorm [37] and SwiGLU activations [56], which are applied consistently across all model sizes and normalization methods. For optimization, we use the Adam

Table 2: Fine-tuning performance (↑) of LLaMA with various layer normalizations.

| Method | MMLU | BoolQ | ARC-e | PIQA | Hellaswag | OBQA | Winogrande | Average |
|---|---|---|---|---|---|---|---|---|
| **LLaMA-250M** | | | | | | | | |
| Post-LN [2] | 22.95 | 37.83 | 26.94 | 52.72 | 26.17 | 11.60 | 49.56 | 32.54 |
| DeepNorm [47] | 23.60 | 37.86 | 36.62 | 61.10 | 25.69 | 15.00 | 49.57 | 35.63 |
| Mix-LN [26] | 26.53 | 56.12 | 41.68 | 66.34 | 30.16 | 18.00 | 50.56 | 41.34 |
| Pre-LN [3] | 24.93 | 38.35 | 40.15 | 63.55 | 26.34 | 16.20 | 49.01 | 36.93 |
| Pre-LN + LayerNorm Scaling | **27.08** | **58.17** | **45.24** | **67.38** | **32.81** | **18.80** | **52.49** | **43.14** |
| **LLaMA-1B** | | | | | | | | |
| Post-LN [2] | 22.95 | 37.82 | 25.08 | 49.51 | 25.04 | 13.80 | 49.57 | 31.96 |
| DeepNorm [47] | 23.35 | 37.83 | 27.06 | 52.94 | 26.19 | 11.80 | 49.49 | 32.67 |
| Mix-LN [26] | 23.19 | 37.83 | 25.08 | 49.51 | 25.04 | 11.80 | 49.57 | 31.72 |
| Pre-LN [3] | 26.54 | **62.20** | 45.70 | 67.79 | 30.96 | 17.40 | 50.51 | 43.01 |
| Pre-LN + LayerNorm Scaling | **28.69** | 61.80 | **48.85** | **67.92** | **33.94** | **18.60** | **54.30** | **44.87** |

optimizer [22] and adopt size-specific learning rates: $1 \times 10^{-3}$ for models up to 350M parameters, and $5 \times 10^{-4}$ for the 1B parameter model. All models share the same architecture, hyperparameters, and training schedule, with the only difference being the choice of normalization method. Unlike Mix-LN [26], which introduces an additional hyperparameter $\alpha$ manually set to 0.25, LayerNorm Scaling requires no extra hyperparameters, making it simpler to implement. Table 1 shows that LayerNorm Scaling consistently outperforms other normalization methods across different model sizes. While DeepNorm performs comparably to Pre-LN on smaller models, it struggles with larger architectures like LLaMA-1B, showing signs of instability and divergence in loss values. Similarly, Mix-LN outperforms Pre-LN in smaller models but faces convergence issues with LLaMA-350M, indicating its sensitivity to architecture design and hyperparameter tuning due to the introduction of Post-LN. Notably, Mix-LN was originally evaluated on LLaMA-1B with 50K steps [26], while our setting extends training to 100K steps, where Mix-LN fails to converge, highlighting its instability in large-scale settings caused by the usage of Post-LN.

In contrast, LayerNorm Scaling solves the *Curse of Depth* without compromising the training stability. LayerNorm Scaling achieves the lowest perplexity across all tested model sizes, showing stable performance improvements over existing methods. For instance, on LLaMA-130M and LLaMA-1B, LayerNorm Scaling reduces perplexity by 0.97 and 1.31, respectively, compared to Pre-LN. Notably, LayerNorm Scaling maintains stable training dynamics for LLaMA-1B, a model size where Mix-LN fails to converge. These findings demonstrate that LayerNorm Scaling provides a robust and computationally efficient normalization strategy, enhancing large-scale training of language models without additional implementation complexity.

## 5.2 Supervised Fine-tuning

To verify whether the gains in pre-training can be translated to the stage of post-training, we perform SFT with the models obtained from Section 5.1 on the Commonsense170K dataset [18] across eight downstream tasks. We adopt the same fine-tuning configurations as used in Li et al. [26]. The results, presented in Table 2, demonstrate that LayerNorm Scaling consistently surpasses other normalization techniques in all evaluated datasets. For the LLaMA-250M model, LayerNorm Scaling improves average performance by 1.80% and achieves a 3.56% gain on ARC-e compared to Mix-LN. Similar trends are observed with the LLaMA-1B model, where LayerNorm Scaling outperforms Pre-LN, Post-LN, Mix-LN, and DeepNorm on seven out of eight tasks, with an average gain of 1.86% over the best baseline. These results confirm that LayerNorm Scaling enhances generalization on diverse downstream tasks by improving the representation quality of deep layers.

## 5.3 Scaling Up Training

### 5.3.1 OLMo

**Model Size Scaling.** To further assess the scalability and robustness of LNS, we conduct additional experiments using the OLMo repository [15], scaling training across model sizes of 60M, 150M, 300M, 1B, and 7B parameters. All models are trained on a fixed 20B-token budget to ensure comparability. These experiments are designed to evaluate whether the performance gains observed with LNS in smaller-scale settings extend to more challenging and state-of-the-art LLM training regimes. As shown in Figure 4, LNS consistently and substantially outperforms the standard Pre-LN

baseline across all model sizes. Remarkably, for the 7B model, LNS reduces the final loss from 2.69 to 2.50. These results underscore the scalability of LNS and its effectiveness in large-scale pre-training scenarios.

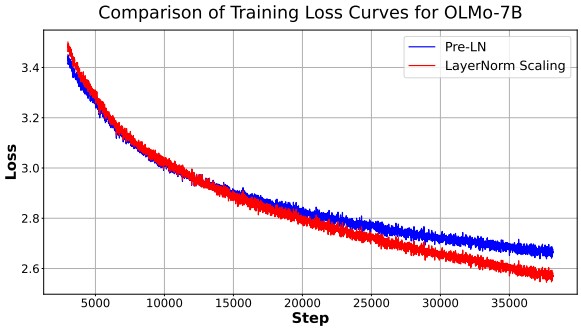

Figure 5: Training loss of OLMo-7B with Pre-LN and LNS.

**Loss Curve.** Figure 5 shows the training loss curves of 7B models trained with Pre-LN and LNS. While LayerNorm Scaling exhibits slightly slower convergence at the early stages of training, it consistently outperforms Pre-LN as training progresses, ultimately achieving a substantial loss gap. We attribute this to the uncontrolled accumulation of output variance in Pre-LN, which amplifies with depth and training steps, ultimately impairing the effective learning of deeper layers. In contrast, LNS mitigates this issue by scaling down the output variance in proportion to depth, thereby enabling more stable and effective training across all layers during training.

**Beating OLMo's Scaled Initialization.** OLMo adopts the scaled initialization proposed in Zhang et al. [57] and used by Mehta et al. [31], which scales input projections by $1/\sqrt{d_{\mathrm{model}}}$, and output projections by $1/\sqrt{2 \cdot d_{\mathrm{model}} \cdot l}$ at every layer. This method is designed to enhance training stability and to scale down variance at initialization. To evaluate the effectiveness of LNS, we compare it against this state-of-the-art initialization by training OLMo-1B on 20B tokens. As shown in Table 3, LNS achieves consistently lower training loss, indicating that it may offer a more effective alternative for large-scale LLM training.

Table 3: Comparison with OLMo's Scaled Initialization.

| Method | Model | # Tokens | Training Loss | Perplexity |
|---|---|---|---|---|
| OLMo's Scaled Initialization | OLMo-1B | 20B | 2.96 | 19.30 |
| LayerNorm Scaling | OLMo-1B | 20B | **2.85** | **17.28** |

### 5.4 LNS Effectively Scales Down Output Variance

As LNS is proposed to reduce output variance, we empirically validate this claim during the pre-training of LLMs. We compare the layerwise output variance of three configurations: (1) the standard Pre-LN [2], (2) Pre-LN with Scaled Initialization [38, 35], which scales the initialization of the feedforward layers' weights $W_0$ and $W_2$ by $\frac{1}{\sqrt{2L}}$, where $L$ is the total number of Transformer layers, and (3) Pre-LN with LNS. The average output variance across layers is shown in Figure 1. For both vanilla Pre-LN and Scaled Initialization, the output variance in shallow layers (blue) remains relatively stable throughout training, while variance in deeper layers (red) grows substantially after 2K iterations, reaching up to 175 in the final layer. Since Scaled Initialization only operates at initialization, it is insufficient to constrain output variance during training. In contrast, LNS consistently suppresses the growth of output variance in deeper layers, capping it at approximately 25.

### 5.5 LNS Enhances the Effectiveness of Deep Layers

Furthermore, to assess whether LNS enhances the effectiveness of deeper layers by promoting more diverse feature representations, we analyze the layerwise performance drop and the angular distance of LNS, as shown in Figure 6. Compared to Pre-LN, the performance degradation in LayerNorm Scaling is more uniformly distributed across layers, indicating a more balanced contribution from

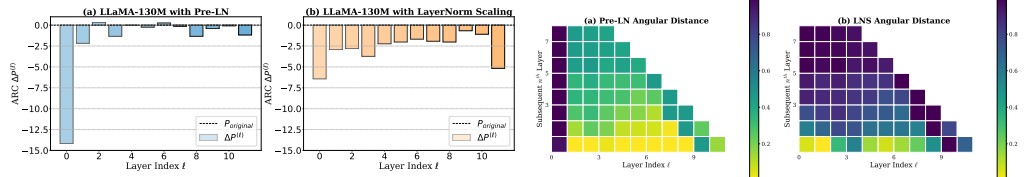

Figure 6: **Left:** Performance drop of layer pruning on LLaMA-130M. **Right:** The angular distance between representations of subsequent layers is shown. LayerNorm Scaling enables deep layers to make a meaningful contribution to the model.

each layer. Notably, pruning the deeper layers of LNS results in a more significant accuracy drop, suggesting these layers play a more critical role in task performance. Additionally, features learned under LNS exhibit greater distinction: most layers show a substantial angular distance, exceeding 0.6, from their adjacent layers, indicating more diverse representations. In sharp contrast, the layerwise angular distance in Pre-LN remains significantly lower and progressively decreases with depth, suggesting reduced feature diversity. We provide studies CoD in Vision Transformers and Vision Language Models in Appendix H and Appendix G, respectively.

## 6 Ablation Study

**Comparing Against Other Scaling Methods.** We first compare LNS with previous scaling approaches, including (1) Scaled Initialization [38, 35], which scales the initialization of $W_0$ and $W_2$ by the overall depth $1/\sqrt{2L}$; (2) Depth-Scaled Initialization [57] scales the initialization of weight matrices by the current depth $1/\sqrt{2l}$; (3) SkipInit [9] introduces a learnable parameter after FFN/Att layers, initialized as $1/\sqrt{L}$; (4) LayerScale [43] applies per-channel weighting using a diagonal matrix, $diag(\lambda_1, \ldots, \lambda_d)$, where each weight $\lambda_i$ is initialized to a small value (e.g., $\lambda_i = \epsilon$). Table 4 presents the results of LLaMA-130M and LLaMA-250M.

First, we observe that methods involving learnable parameters, such as LayerScale and SkipInit, consistently degrade performance in LLMs. Among initialization-based techniques, a larger scaling factor proves beneficial: Scaled Initialization yields lower perplexity compared to Depth-Scaled Initialization. Notably, LNS achieves the best overall performance, underscoring the advantage of applying scaling dynamically during training. **Interestingly**, combining LNS with Scaled Initialization results in worse performance than using LNS alone, highlighting the importance of removing conflicting initialization strategies prior to adopting LNS.

Table 4: Comparing LNS against other scaling methods. Perplexity ($\downarrow$) is reported.

|  | LLaMA-130M | LLaMA-250M |
|---|---|---|
| Training Tokens | 2.2B | 3.9B |
| Pre-LN | 26.73 | 21.92 |
| + LayerScale [43] | 27.93 | 23.45 |
| + SkipInit [9] | 27.41 | 22.29 |
| + Depth-Scaled Initialization [57] | 26.95 | 21.50 |
| + Scaled Initialization [38] | 26.04 | 20.98 |
| + LayerNorm Scaling | **25.76** | **20.35** |
| + LayerNorm Scaling + Scaled Initialization | 25.80 | 20.79 |

**Comparison with Other Layer Normalization.** In addition, we conducted comparisons using LLaMA-130M to evaluate LayerNorm Scaling against recently proposed normalization methods, including Admin [28], Sandwich-LN [11], Group-LN [50, 30], and Mix-LN [26]. Table 5 shows that Admin and Group-LN degrade performance. Sandwich-LN slightly outperforms Pre-LN. Both Mix-LN and LayerNorm Scaling improve over Pre-LN by good margins. However, Mix-LN fails to reduce perplexity under 26, falling short of LayerNorm Scaling and suffers from instability in large-scale scenarios as shown in Table 1.

Table 5: Comparison against other normalization methods on LLaMA-130M. All methods use the identical configurations. Perplexity ($\downarrow$) is reported.

| Pre-LN | Admin | Group-LN | Sandwich-LN | Mix-LN | LayerNorm Scaling |
|---|---|---|---|---|---|
| 26.73 | 27.91 | 28.01 | 26.51 | 26.07 | **25.76** |

**Effect of Positions of LNS.** The results in Table 6 show that inserting the scaling factor at different points can have a considerable influence on the model's performance. Placing it after the residual connection ("After Residual") leads to a perplexity of 1358.11, which indicates training divergence. In contrast, LNS incorporates the scaling factor after LN achieving the best perplexity of 25.76, surpassing both the baseline Pre-LN setting (26.73) and other placements. This suggests that modifying the LayerNorm to include the scaling factor can enhance training stability and final performance for this model configuration.

Table 6: Comparison against different scaling factor position on LLaMA-130M.

| Pre-LN | Before LN | After Attn/FFN | After Residual | LNS Only After Attn | LNS Only After FFN | Ours (After LN) |
|---|---|---|---|---|---|---|
| 26.73 | 26.97 | 26.53 | 1358.11 | 26.89 | 26.43 | **25.76** |

# 7   Related Work

**Ineffectiveness of Deeper Layers in Transformers.** The ineffectiveness of deep layers in LLMs has been previously reported. Yin et al. [54] found that deeper layers of LLMs can tolerate significantly higher levels of pruning compared to shallower layers, achieving high sparsity. Similarly, Gromov et al. [16] and Men et al. [32] demonstrated that removing early layers causes a dramatic decline in model performance, whereas removing deep layers does not. Lad et al. [23] showed that the middle and deep layers of GPT-2 and Pythia exhibit remarkable robustness to perturbations such as layer swapping and layer dropping. Recently, Li et al. [25] highlighted that early layers contain more outliers and are therefore more critical for fine-tuning. While these studies effectively highlight the limitations of deep layers in LLMs, they stop short of identifying the root cause of this issue or proposing viable solutions to address it.

**Layer Normalization in Language Models.** LN [2] was initially applied after the residual connection in the original Transformer [45], which is known as Post-LN. Later on, Pre-LN [3, 8, 34] dominated LLMs, due to its compelling performance and stability [7, 44, 21, 6]. Prior works have studied the effect of Pre-LN and Post-LN. Xiong et al. [51] proves that Post-LN tends to have larger gradients near the output layer, which necessitates smaller learning rates to stabilize training, whereas Pre-LN scales down gradients with the depth of the model, working better for deep Transformers. Wang et al. [48] empirically confirmed that Pre-LN facilitates stacking more layers and Post-LN suffers from gradient vanishing. The idea of connecting multiple layers was proposed in previous works [5, 12, 48]. Admin introduces additional parameters to control residual dependencies, stabilizing Post-LN. DeepNorm [47] enables stacking 1000-layer Transformers by upscaling the residual connection before applying LN. Additionally, Ding et al. [11] proposed Sandwich LayerNorm, normalizing both the input and output of each transformer sub-layer. Takase et al. [40] introduced B2T to bypass all LN except the final one in each layer. Li et al. [26] recently combines Post-LN and Pre-LN to enhance the middle layers. Zhu et al. [60] introduces Dynamic Tanh (DyT) as a normalization-free alternative in Transformers, delivering comparable performance. Zhuo et al. [61] proposes HybridNorm, a hybrid normalization scheme combining QKV normalization with Post-Norm FFN to stabilize training in deep transformers. De and Smith [9] also states that normalized residual blocks in deep networks are close to the identity function and proposes SkipInit to remove normalization by introducing a learnable scalar multiplier on the residual branch initialized to $1/\sqrt{L}$. Our experiments suggest that SkipInit's learnable parameter does not improve performance and sometimes harms training.

# 8   Conclusion

In this paper, we re-introduce the concept of the *Curse of Depth* in LLMs, highlighting an urgent yet often overlooked phenomenon: nearly half of the deep layers in modern LLMs are less effective than expected. We discover the root cause of this phenomenon is Pre-LN which is widely used in almost all modern LLMs. To tackle this issue, we introduce *LayerNorm Scaling*. By scaling the output variance inversely with the layer depth, LayerNorm Scaling ensures that all layers, including deeper ones, contribute meaningfully to training. Our experiments show that this simple modification improves performance, reduces resource usage, and stabilizes training across various model sizes. LayerNorm Scaling is easy to implement, hyperparameter-free, and provides a robust solution to enhance the efficiency and effectiveness of LLMs.

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

# A   Proofs of the Theorems of curse of depth

## A.1   Proof of Lemma 1

*Proof.* Given Equation (2) from [41], we have:

$$y = x_{\ell+1} = x'_\ell + \text{FFN}(\text{LN}(x'_\ell)),$$
$$x'_\ell = x_\ell + \text{Attn}(\text{LN}(x_\ell)). \tag{13}$$

Based on our Assumption 1, let $\text{Var}(\text{Attn}(\text{LN}(x_\ell))) = \sigma^2_{\text{Attn}}$. Then we can write:

$$\text{Var}(x'_\ell) = \text{Var}(x_\ell) + \text{Var}(\text{Attn}(\text{LN}(x_\ell))) + \text{Cov}(\text{Attn}(\text{LN}(x_\ell)), \text{Var}(x_\ell))$$
$$= \sigma^2_{x_\ell} + \sigma^2_{\text{Attn}} + \rho_1 \cdot \sigma_{x_\ell} \cdot \sigma_{\text{Attn}}, \tag{14}$$

where $\rho_1$ is the correlation factor. Similarly, let $\text{Var}(\text{FFN}(\text{LN}(x'_\ell))) = \sigma^2_{\text{FFN}}$. Then we have:

$$\sigma^2_{x_{\ell+1}} = \sigma_{(x'_\ell)^2} + \sigma^2_{\text{FFN}} + \rho_2 \cdot \sigma_{x'_\ell} \cdot \sigma_{\text{FFN}}, \tag{15}$$

where $\rho_2$ is the correlation factor. Thus, the relationship between $\text{Var}(x_{\ell+1})$ and $\text{Var}(x_\ell)$ becomes:

$$\sigma^2_{x_{\ell+1}} = \sigma^2_{x_\ell} + \sigma^2_{\text{Attn}} + \sigma^2_{\text{FFN}} + \rho_1 \cdot \sigma_{x_\ell} \cdot \sigma_{\text{Attn}} + \rho_2 \cdot \sigma_{x'_\ell} \cdot \sigma_{\text{FFN}}. \tag{16}$$

### A.1.1   Variance of the Attention

The scaled dot-product attention mechanism is defined as:

$$\text{Attn}(Q, K, V) = \text{softmax}\left(\frac{QK^T}{\sqrt{d_k}}\right) V. \tag{17}$$

The softmax function outputs a probability distribution over the keys. Let the softmax output be $A = \text{softmax}\left(\frac{QK^T}{\sqrt{d_k}}\right)$, where $A$ is a matrix with each row summing to 1. The final attention output is obtained by multiplying the softmax output $A$ with the value matrix $V$:

$$\text{Attn}(Q, K, V) = AV. \tag{18}$$

**Lemma 2** ([24]). *Let $\{X_i\}_{i=1}^N$ be independent and identically distributed random variables with mean $m$ and variance $\sigma^2 < \infty$. Define the softmax weights $p_i = \frac{e^{X_i}}{\sum_{j=1}^N e^{X_j}}$, and let $p = (p_1, \ldots, p_N)$. Then, as $N \to \infty$, with high probability, the softmax vector $p$ concentrates around the uniform distribution on $N$ elements. In particular,*

$$\lim_{N \to \infty} \mathbb{E}\left[\sum_{i=1}^N \left(p_i - \frac{1}{N}\right)^2\right] = 0, \tag{19}$$

*which implies that the softmax output becomes asymptotically indistinguishable, in expectation, from the uniform distribution.*

According to the above lemma, to simplify the analysis, we make the following additional assumptions: The softmax output $A$ is approximately uniform, meaning each element of $A$ is roughly $1/n$, where $n$ is the number of keys/values. Given this assumption, the variance of the attention is:

$$\text{Var}(\text{Attn}(Q, K, V)) \sim \text{Var}(AV) = \frac{1}{n} \sum_{i=1}^n d_{\text{head}} \text{Var}(V_i) = \frac{1}{n} \cdot n\sigma_V^2 \cdot d_{\text{head}} = d_{\text{head}}\sigma_V^2 = \sigma_W^2 d. \tag{20}$$

where $W$ is the universal weight matrix defined as before.

### A.1.2 Variance of the Feed-Forward Network

The feed-forward network (FFN) in transformers typically consists of two linear transformations with a ReLU activation in between. The FFN can be written as:

$$\text{FFN}(x) = W_2 \cdot \text{ReLU}(W_1 \cdot x + b_1) + b_2. \tag{21}$$

where $W_1$ and $W_2$ are weight matrices, and $b_1$ and $b_2$ are bias vectors.

Using the result obtained by Wang et al. [47], we get:

$$\sigma^2_{\text{FFN}} \sim \sigma^2_{W_1} \cdot \sigma^2_{W_2} = \sigma^4_W. \tag{22}$$

In conclusion:

$$
\begin{aligned}
\sigma^2_{x'_\ell} &= \sigma^2_{x_\ell} + \sigma^2_W + \rho_2 \cdot \sigma_{x_\ell} \cdot \sigma_W \\
&= \sigma^2_{x_\ell}\left(1 + \frac{\sigma_W}{\sigma_{x_\ell}} + \frac{\sigma^2_W}{\sigma^2_{x_\ell}}\right) \\
&= \sigma^2_{x_\ell}\Theta\left(1 + \frac{1}{\sigma_{x_\ell}}\right).
\end{aligned} \tag{23}
$$

For simplicity, we set the numerator part to 1. Substitute $\sigma_{x'_\ell} = \sigma_{x_\ell}\sqrt{1 + \frac{\sigma^2_W}{\sigma^2_{x_\ell}} + \rho_2 \cdot \frac{\sigma_W}{\sigma_{x_\ell}}}$. into Equation (16) we get:

$$
\begin{aligned}
\sigma^2_{x_{\ell+1}} &= \sigma^2_{x_\ell} + \sigma^2_W + \sigma^4_W d^2 + \rho_1 \cdot \sigma_{x_\ell} \cdot \sigma_W + \rho_2 \cdot \sigma_{x'_\ell} \cdot \sigma^2_W d \\
&= \sigma^2_{x_\ell} + \sigma^2_W + \sigma^4_W d^2 + \rho_1 \cdot \sigma_{x_\ell} \cdot \sigma_W + \rho_2 \cdot \sigma_{x_\ell} \cdot \sigma^2_W d + \frac{\rho_2 \sigma^4_W d^2}{2\sigma_{x_\ell}} + \frac{\rho^2_2 \sigma^3_W d \sigma_{x_\ell}}{2} \\
&= \sigma^2_{x_\ell}\Theta\left(1 + \frac{1}{\sigma_{x_\ell}}\right).
\end{aligned} \tag{24}
$$

From the result we can generally infer that the variance accumulates layer by layer. The variance with regard to $\sigma_{x_1}$:

$$\sigma^2_{x_\ell} = \sigma^2_{x_1}\Theta\left(\prod_{k=1}^{\ell-1}\left(1 + \frac{1}{\sigma_{x_k}}\right)\right). \tag{25}$$

We can also obtain a similar result for $\sigma^2_{x'_\ell}$.

We observe that for any $\sigma^2_{x_k} \geq 1$, the sequence is increasing, meaning each term in the product is bounded. Consequently, the entire product is bounded above by:

$$\sigma^2_{x_\ell} \leq \sigma^2_{x_1}\prod_{k=1}^{\ell-1}\left(1 + \sqrt{\frac{1}{\sigma_{x_1}}}\right) = \sigma^2_{x_1}\left(1 + \sqrt{\frac{1}{\sigma_{x_1}}}\right)^{\ell-1} = \exp\Theta(L). \tag{26}$$

Taking the natural logarithm of both sides:

$$
\begin{aligned}
\log(\sigma^2_{x_\ell}) &= \log\left(\sigma^2_{x_1}\prod_{k=1}^{\ell-1}\left(1 + \sqrt{\frac{1}{\sigma^2_{x_k}}}\right)\right) = \sum_{k=1}^{\ell-1}\log\left(1 + \sqrt{\frac{1}{\sigma^2_{x_k}}}\right) + \log(\sigma^2_{x_1}) \\
&\geq \sum_{k=1}^{\ell-1}\left(\sqrt{\frac{1}{\sigma^2_{x_k}}} - \frac{1}{2}\left(\sqrt{\frac{1}{\sigma^2_{x_k}}}\right)^2\right) + \log(\sigma^2_{x_1}).
\end{aligned} \tag{27}
$$

Exponentiating both sides to find the lower bound for $\sigma^2_{x_\ell}$, we obtain:

$$\sigma^2_{x_\ell} \geq \sigma^2_{x_1}\exp\left(\sum_{k=1}^{\ell-1}\left(\sqrt{\frac{1}{\sigma^2_{x_k}}} - \frac{1}{2\sigma^2_{x_k}}\right)\right).$$

This provides a tighter lower bound for $\sigma_{x_\ell}^2$ compared to the upper bound of Equation (26). Since we know the upper bound of variance grows exponentially, the lower bound must be sub-exponential. Therefore, for $\sigma_{x_\ell}^2 = \ell$, we must have:

$$\sigma_{x_\ell}^2 \geq \sigma_{x_1}^2 \exp\left(\sum_{k=1}^{\ell-1}\left(\frac{1}{k}-\frac{1}{2k}\right)\right) = \Theta(\exp(\sqrt{L})) \geq \Theta(L).$$

$\square$

Therefore, the increasing lower bound for $\sigma_{x_\ell}^2$ must grow faster than a linear function. So, the increase of variance is sub-exponential. A large increase in such bound will lead to gradient spikes, which can connect to previous studies in Huang et al. [19, 20].

## A.2 Proof of Theorem 1

In this proof, we will divide the argument into two parts: first, the calculation of the Lemma 3, and second, the analysis of $\frac{\partial y_\ell}{\partial x_1}$.

**Lemma 3.** *For an L-layered Pre-LN Transformer, $\frac{\partial y_L}{\partial x_1}$ using Equations (2) and (3) is given by:*

$$\frac{\partial y_L}{\partial x_1} = \prod_{n=1}^{L-1}\left(\frac{\partial y_\ell}{\partial x'_\ell}\cdot\frac{\partial x'_\ell}{\partial x_\ell}\right). \tag{28}$$

*The upper bound for the norm of $\frac{\partial y_L}{\partial x_1}$ is:*

$$\left\|\frac{\partial y_L}{\partial x_1}\right\|_2 \leq \prod_{l=1}^{L-1}\left(\left(1+\frac{\sigma^2}{\sigma_{x'_\ell}(\sqrt{d}+\sqrt{d_{\mathrm{FFN}}})^2}\right)\right.$$
$$\left.\times\left(1+2dh\left(\sqrt{s}+2+\frac{1}{\sqrt{s}}\right)\frac{\sigma^2}{\sigma_{x_\ell}}\left(\sigma^2 d\sqrt{d_{\mathrm{head}}}+\left(1+\sqrt{d_{\mathrm{head}}/d}\right)\right)\right)\right). \tag{29}$$

Here, $h$ denotes the number of heads, $s$ is the sequence length, and $d$, $d_{\mathrm{FFN}}$, and $d_{\mathrm{head}}$ are the dimension of the embedding, FFN layer and multi-head attention layer, respectively. The standard deviation of $W_Q$, $W_K$, $W_V$, and $W_{\mathrm{FFN}}$ at layer $\ell$ is $\sigma$ based on Assumption 1.

### A.2.1 Proof of Lemma 3

*Proof.* Our derivation follows results in [41], specifically Equation (7), which provides an upper bound on the norm of $\frac{\partial y_\ell}{\partial x_1}$ as:

$$\left\|\frac{\partial y_\ell}{\partial x_1}\right\|_2 = \left\|\prod_{l=1}^{L-1}\frac{\partial y_\ell}{\partial x'_\ell}\frac{\partial x'_\ell}{\partial x_\ell}\right\|_2. \tag{30}$$

Thus, we can estimate the upper bound of the gradient norm of $\frac{\partial y_\ell}{\partial x_1}$ by analyzing the spectral norms of the Jacobian matrices for the FFN layer and the self-attention layer, namely,

$$\mathrm{FFN:}\left\|\frac{\partial y_\ell}{\partial x'_\ell}\right\|_2 \quad \mathrm{Attention:}\left\|\frac{\partial x'_\ell}{\partial x_\ell}\right\|_2. \tag{31}$$

We now derive an upper bound of $\|\frac{\partial y_\ell}{\partial x'_\ell}\|_2$ as follows:

$$\left\|\frac{\partial y_\ell}{\partial x'_\ell}\right\|_2 \leq 1+\left\|\frac{\partial\mathrm{FFN}(\mathrm{LN}(x'_\ell))}{\partial\mathrm{LN}(x'_\ell)}\right\|_2\left\|\frac{\partial\mathrm{LN}(x'_\ell)}{\partial x'_\ell}\right\|_2. \tag{32}$$

Let $\sigma_{w1_\ell}$ and $\sigma_{w2_\ell}$ be the standard deviations of $W_\ell^1$ and $W_\ell^2$, respectively. From Assumption 1, the spectral norms of $W_\ell^1$ and $W_\ell^2$ are given by their standard deviations and dimensions [46], so we have:

$$\|W_1\|_2 \sim \sigma_1(\sqrt{d}+\sqrt{d_{\mathrm{FFN}}}). \tag{33}$$

For simplicity, we assume that $d$, and $d_{\text{FFN}}$ are equal, thus,

$$\left\|\frac{\partial \text{FFN}(\text{LN}(x'_\ell))}{\partial \text{LN}(x'_\ell)}\right\|_2 = \|W_\ell^1 W_\ell^2\|_2 \le \sigma_1 \sigma_2 (\sqrt{d} + \sqrt{d_{\text{ffn}}})^2. \tag{34}$$

Finally, we have the following bound:

$$\left\|\frac{\partial y_\ell}{\partial x'_\ell}\right\|_2 \le 1 + \frac{\sigma_{w1_\ell}\sigma_{w2_\ell}(\sqrt{d}+\sqrt{d_{\text{FFN}}})^2}{\sigma_{x'_\ell}} = 1 + \frac{\sigma_\ell^2(\sqrt{d}+\sqrt{d_{\text{FFN}}})^2}{\sigma_{x'_\ell}}. \tag{35}$$

Following a similar procedure for the FFN, we rewrite $\|\frac{\partial x'}{\partial x}\|_2$ in Equation (31) as:

$$\left\|\frac{\partial x'}{\partial x}\right\|_2 \le 1 + \left\|\frac{\partial \text{Attn}(\text{LN}(x))}{\partial \text{LN}(x)}\right\|_2 \left\|\frac{\partial \text{LN}(x)}{\partial x}\right\|_2. \tag{36}$$

Let $Z(\cdot) = \text{concat}(\text{head}_1(\cdot), \dots, \text{head}_h(\cdot))$ and $J^Z$ denote the Jacobian of the $Z(\cdot)$. We can now express the spectral norm of the Jacobian matrix of attntion as:

$$\left\|\frac{\partial \text{Attn}(\text{LN}(x_\ell))}{\partial \text{LN}(x_\ell)}\right\|_2 = \left\|W_\ell^O Z(\text{LN}(x_\ell))\frac{\partial Z(\text{LN}(x_\ell))}{\partial \text{LN}(x_\ell)}\right\|_2 = \|W_\ell^O J_\ell^Z\|_2. \tag{37}$$

From [46], we know that:

$$\|J_\ell^Z\|_2 \le h\left(\left(\sqrt{s}+2+\frac{1}{\sqrt{s}}\right)\sigma^3\sqrt{d^3 d_{\text{head}}} + \sigma_x^\ell\left(\sqrt{d}+\sqrt{d_{\text{head}}}\right)\right). \tag{38}$$

Here $h$ is the number of heads, $s$ is the sequence length, and the standard deviation of $W_Q$, $W_K$, and $W_V$ is $\sigma$.

By combining the inequalities (35), (38) and (36), and assuming that all $\sigma$ values are the same for simplicity. we obtain:

$$\begin{aligned}
\left\|\frac{\partial y_L}{\partial x_1}\right\|_2 \le \prod_{l=1}^{L-1} & \left(\left(1+\frac{\sigma^2(\sqrt{d}+\sqrt{d_{\text{FFN}}})^2}{\sigma_{x'_\ell}}\right)\right. \\
& \left.\times\left(1+2dh\left(\sqrt{s}+2+\frac{1}{\sqrt{s}}\right)\frac{\sigma^2}{\sigma_{x_\ell}}\left(\sigma^2 d\sqrt{d_{\text{head}}}+\left(1+\sqrt{d_{\text{head}}/d}\right)\right)\right)\right).
\end{aligned} \tag{39}$$

$\square$

### A.2.2  Analysis of the Upper Bound

As discussed in [41], $\sigma$ should be sufficiently small, and the standard deviation, $\sigma_{x'_\ell}$ or $\sigma_{x_\ell}$ should satisfy the condition $\sigma^2 \ll \sigma_{x'_\ell}$ to maintain the lazy training scheme. Thus, we obtain the following bound for the product over $\ell$ from 1 to $L$:

To find the bound for $\left\|\frac{\partial y_\ell}{\partial x_1}\right\|_2$ with respect to $\ell$, we simplify the given inequality by approximating $\sigma_{x_\ell}$ and $\sigma_{x'_\ell}$. Based on Equation (23), $\sigma_{x_\ell}$ is only one layer ahead of $\sigma_{x'_\ell}$, and this layer does not significantly affect the overall performance of deep Transformer networks. Furthermore, based on Lemma 1, we assume that $\sigma_{x'_\ell} = \sigma_{x_\ell}$.

Equation (3) can be expressed in a traditional product form [49] for $\sigma_{x_\ell}$:

$$\left\|\frac{\partial y_L}{\partial x_1}\right\|_2 \le \prod_{l=1}^{L-1}\left(1+\frac{1}{\sigma_{x_\ell}}A+\frac{1}{\sigma_{x_\ell}^2}B\right), \tag{40}$$

where

$$A = \frac{\sigma^2}{(\sqrt{d}+\sqrt{d_{\text{FFN}}})^2}+2dh\left(\sqrt{s}+2+\frac{1}{\sqrt{s}}\right)\sigma^2\left(d\sqrt{d_{\text{head}}}+1+\sqrt{d_{\text{head}}/d}\right), \tag{41}$$

and

$$B = 2dh \left( \sqrt{s} + 2 + \frac{1}{\sqrt{s}} \right) \sigma^4 d \sqrt{d_{\text{head}}}, \tag{42}$$

where $A$ and $B$ are independent of $\sigma_{x_\ell}$, and under our assumption, are treated as constants.

From classical infinite series analysis, it is known that as $\sigma_{x_\ell}$ grows at a faster rate, the upper bound of the product decreases. The proof is omitted here for brevity. For the upper bound on the convergence rate of $\sigma_{x_\ell}^2$, we assume $\sigma_{x_\ell}^2 = \exp(\ell)$ without loss of generality. Under this condition, we can derive the following result:

Taking the natural logarithm of the product:

$$\log \left( \prod_{k=1}^{L-1} \left( 1 + \frac{A}{e^k} + \frac{B}{e^{2k}} \right) \right) = \sum_{k=1}^{L-1} \log \left( 1 + \frac{A}{e^k} + \frac{B}{e^{2k}} \right).$$

Using the Taylor series expansion for $\log(1 + x)$, and applying this to our sum, we get:

$$\sum_{k=1}^{\infty} \log \left( 1 + \frac{A}{e^k} + \frac{B}{e^{2k}} \right) = \sum_{k=1}^{\infty} \left( \frac{A}{e^k} + \frac{B}{e^{2k}} - \frac{1}{2} \left( \frac{A}{e^k} + \frac{B}{e^{2k}} \right)^2 + \frac{1}{3} \left( \frac{A}{e^k} + \frac{B}{e^{2k}} \right)^3 - \cdots \right).$$

By evaluating the sums for each order of terms, we find that the result is a constant. Carrying this out for each term, we obtain:

$$\log \left( \prod_{k=1}^{L-1} \left( 1 + \frac{A}{e^k} + \frac{B}{e^{2k}} \right) \right) \sim \frac{A}{e-1} + \frac{B}{e^2-1} - \frac{1}{2} \left( \frac{A^2}{e^2-1} + 2\frac{A \cdot B}{e^3-1} + \frac{B^2}{e^4-1} \right).$$

Thus, the product is approximately:

$$\left\| \frac{\partial y_L}{\partial x_1} \right\|_2 \leq \exp \left( \frac{A}{e-1} + \frac{B}{e^2-1} - \frac{1}{2} \left( \frac{A^2}{e^2-1} + 2\frac{A \cdot B}{e^3-1} + \frac{B^2}{e^4-1} \right) \right) = M, \tag{43}$$

where $M$ is a constant.

For the lower bound on the convergence rate of $\sigma_{x_\ell}^2$, we assume $\sigma_{x_\ell}^2 = \ell$ without loss of generality. Under this condition, we derive the following result. Taking the logarithm of the product, applying the Taylor series expansion for $\log(1 + x)$, and applying this to our sum:

$$\sum_{k=1}^{\infty} \log \left( 1 + \frac{A}{k} + \frac{B}{e^{k^2}} \right) = \sum_{k=1}^{\infty} \left( \frac{A}{k} + \frac{B}{e^{k^2}} - \frac{1}{2} \left( \frac{A}{k} + \frac{B}{e^{k^2}} \right)^2 + \frac{1}{3} \left( \frac{A}{k} + \frac{B}{e^{k^2}} \right)^3 - \cdots \right).$$

For the first-order terms:

$$\sum_{k=1}^{\infty} \left( \frac{A}{k} + \frac{B}{e^{k^2}} \right) = A \sum_{k=1}^{\infty} \frac{1}{k} + B \sum_{k=1}^{\infty} \frac{1}{e^{k^2}}.$$

The series $\sum_{k=1}^{\infty} \frac{1}{k}$ is the harmonic series, which diverges. However, we approximate it using the Euler-Mascheroni constant $\gamma$ and the fact recognize that the harmonic series grows logarithmically:

$$\sum_{k=1}^{\infty} \frac{1}{k} \sim \log n + \gamma \quad \text{(for large } n\text{)}.$$

The other series such as $\sum_{k=1}^{\infty} \frac{1}{e^{k^2}}$ converge because $e^{k^2}$ grows very rapidly.

For higher-order terms, they converge to constant, involving the series $\sum_{k=1}^{\infty} \frac{1}{k^2}$ converges to $\frac{\pi^2}{6}$, so they contribute a constant. Exponentiating both sides, we get:

$$\prod_{k=1}^{\infty} \left( 1 + \frac{A}{k} + \frac{B}{e^{k^2}} \right) \sim \exp\left( A(\log n + \gamma) + const \right).$$

Thus, the growth rate of the upper bound for $\left\| \frac{\partial y_L}{\partial x_1} \right\|_2$ is:

$$\left\| \frac{\partial y_L}{\partial x_1} \right\|_2 \leq \Theta(L). \tag{44}$$

## B    Theoretical Analysis of LayerNorm Scaling

**Lemma 4.** *After applying our scaling method, the variances of $x'_\ell$ and $x_\ell$, denoted as $\sigma^2_{x'_\ell}$ and $\sigma^2_{x_\ell}$, respectively, exhibit the same growth trend, which is:*

$$\sigma^2_{x_\ell} = \sigma^2_{x_1} \Theta\Big( \prod_{k=1}^{\ell-1} \Big( 1 + \frac{1}{\sqrt{k}\sigma_{x_k}} \Big) \Big), \tag{45}$$

*with the following growth rate bounds:*

$$\Theta(L) \leq \sigma^2_{x_L} \leq \Theta(L^{(2-\epsilon)}). \tag{46}$$

*where $\epsilon$ is a small number with $1/2 \leq \epsilon < 1$.*

From Lemma 4, we can conclude that our scaling method effectively slows the growth of the variance upper bound, reducing it from exponential to polynomial growth. Specifically, it limits the upper bound to a quadratic rate instead of an exponential one. Based on Theorem 1, after scaling, we obtain the following:

**Theorem 2.** *For the scaled Pre-LN Transformers, the Euclidean norm of $\frac{\partial y_L}{\partial x_1}$ is given by:*

$$\left\| \frac{\partial y_L}{\partial x_1} \right\|_2 \leq \prod_{\ell=1}^{L-1} \left( 1 + \frac{1}{\ell \sigma_{x_\ell}} A + \frac{1}{\ell^2 \sigma^2_{x_\ell}} B \right), \tag{47}$$

*where $A$ and $B$ are dependent on the scaled neural network parameters. Then the upper bound for the norm is given as follows: when $\sigma^2_{x_\ell}$ grows at $\ell^{(2-\epsilon)}$, (i.e., at its upper bound), we obtain:*

$$\sigma^2_{x_\ell} \sim \ell^{(2-\epsilon)}, \quad \left\| \frac{\partial y_L}{\partial x_1} \right\|_2 \leq \omega(1), \tag{48}$$

*where $\omega$ denotes that if $f(x) = \omega(g(x))$, then $\lim_{x \to \infty} \frac{f(x)}{g(x)} = \infty$. Meanwhile, when $\sigma^2_{x_\ell}$ grows linearly (i.e., at its lower bound), we obtain:*

$$\sigma^2_{x_\ell} \sim \ell, \quad \left\| \frac{\partial y_L}{\partial x_1} \right\|_2 \leq \Theta(L). \tag{49}$$

The detailed descriptions of $A$ and $B$, and $\epsilon$, along with the full proof, are provided in Appendices B.1 and B.2.

By comparing Theorem 1 (before scaling) with Theorem 2 (after scaling), we observe a substantial reduction in the upper bound of variance. Specifically, it decreases from exponential growth $\Theta(\exp(L))$ to at most quadratic growth $\Theta(L^2)$. In fact, this growth is even slower than quadratic expansion, as it follows $\Theta(L^{(2-\epsilon)})$ for some small $\epsilon > 0$.

When we select a reasonable upper bound for this expansion, we find that $\left\| \frac{\partial y_L}{\partial x_1} \right\|_2$ no longer possesses a strict upper bound. That is, as the depth increases, $\left\| \frac{\partial y_L}{\partial x_1} \right\|_2$ continues to grow gradually. Consequently, fewer layers act as identity mappings compared to the original Pre-LN where nearly all deep layers collapsed into identity transformations. Instead, the after-scaled network effectively utilizes more layers, even as the depth approaches infinity, leading to improved expressivity and trainability.

## B.1 Proof of Lemma 4

*Proof.* After scaling, the equation becomes:

$$y = x_{\ell+1} = x'_\ell + \text{FFN}(\frac{1}{\sqrt{\ell}}\text{LN}(x'_\ell)),$$

$$x'_\ell = x_\ell + \text{Attn}(\frac{1}{\sqrt{\ell}}\text{LN}(x_\ell)). \tag{50}$$

Following the same analysis as before, we scale the Attention and FFN sub-layers, yielding:

$$\sigma_{\text{Attn}}^2 = \frac{1}{n\ell} \cdot n \cdot \sigma_V^2 = \frac{1}{\ell}\sigma_V^2 = \frac{\sigma_W^2}{\ell}, \quad \sigma_{\text{FFN}}^2 \sim \frac{\sigma_{W_1}^2}{\ell} \cdot \frac{\sigma_{W_2}^2}{\ell} = \frac{\sigma_W^4}{\ell^2}. \tag{51}$$

In conclusion:

$$\sigma_{x'_\ell}^2 = \sigma_{x_\ell}^2 + \sigma_W^2 + \rho_2 \cdot \sigma_{x_\ell} \cdot \frac{\sigma_W}{\sqrt{\ell}} = \sigma_{x_\ell}^2 \Theta(1 + \frac{1}{\sqrt{\ell}\sigma_{x_\ell}}). \tag{52}$$

Similarly, we obtain:

$$\sigma_{x_{\ell+1}}^2 = \sigma_{x_\ell}^2 \Theta(1 + \frac{1}{\sqrt{\ell}\sigma_{x_\ell}}). \tag{53}$$

From the result we can generally infer that the variance accumulates layer by layer. The variance with regard to $\sigma_{x_1}$:

$$\sigma_{x_\ell}^2 = \sigma_{x_1}^2 \Theta\Big( \prod_{k=1}^{\ell-1} \Big(1 + \frac{1}{\sqrt{k}\sigma_{x_k}}\Big)\Big), \tag{54}$$

We can also obtain a similar result for $\sigma_{x'_\ell}^2$.

Taking the natural logarithm of both sides:

$$\log(\sigma_{x_\ell}^2) = \log\left(\sigma_{x_1}^2 \prod_{k=1}^{\ell-1}\left(1 + \sqrt{\frac{1}{k\sigma_{x_k}^2}}\right)\right) = \sum_{k=1}^{\ell-1}\log\left(1 + \sqrt{\frac{1}{k\sigma_{x_k}^2}}\right) + \log(\sigma_{x_1}^2)$$

$$\geq \sum_{k=1}^{\ell-1}\left(\sqrt{\frac{1}{k\sigma_{x_k}^2}} - \frac{1}{2}\left(\sqrt{\frac{1}{k\sigma_{x_k}^2}}\right)^2\right) + \log(\sigma_{x_1}^2). \tag{55}$$

To establish a lower bound for $\sigma_{x_\ell}^2$, we exponentiate both sides. Setting $\sigma_{x_\ell}^2 = \ell$, we must have:

$$\sigma_{x_\ell}^2 \geq \sigma_{x_1}^2 \exp\left(\sum_{k=1}^{\ell-1}\left(\frac{1}{k} - \frac{1}{2k}\right)\right) = \Theta(\exp(\log L)) \geq \Theta(L). \tag{56}$$

Therefore, the increasing lower bound $\sigma_{x_\ell}^2$ is greater than a linear function.

Similarly, assuming $\sigma_{x_\ell}^2 = \ell^{(2-\epsilon)}$, we have:

$$\sigma_{x_\ell}^2 = \sigma_{x_1}^2 \prod_{k=1}^{\ell-1}\left(1 + \frac{1}{k^{2-\epsilon/2}}\right) \sim \exp\left(\sum_{k=1}^{\ell-1}\frac{1}{k^{2-\epsilon/2}}\right) \sim \exp\left(\frac{\ell^{\epsilon/2-1} - 1}{\epsilon/2 - 1}\right)$$

$$\leq \Theta(\ell^{(2-\epsilon)}) \leq \Theta(\ell^2). \tag{57}$$

Here $\epsilon$ is a small constant with $1/2 \leq \epsilon < 1$. Therefore, the increasing upper bound of $\sigma_{x_\ell}^2$ is slower than the $\ell^3$ function, leading to:

$$\sigma_{x_\ell}^2 \leq \Theta(L^2)$$

.

$\square$

## B.2 Proof of Theorem 2

*Proof.* Similarly, after applying the scaling transformation, we derive an upper bound for $\|\frac{\partial y_\ell}{\partial x'_\ell}\|_2$ as follows:

$$\left\|\frac{\partial y_\ell}{\partial x'_\ell}\right\|_2 \le 1 + \left\|\frac{\partial \mathrm{FFN}(\mathrm{LN}(x'_\ell))}{\partial \mathrm{LN}(x'_\ell)}\right\|_2 \left\|\frac{1}{\sqrt{\ell}}\right\|_2 \left\|\frac{\partial \mathrm{LN}(x'_\ell)}{\partial x'_\ell}\right\|_2$$

$$= 1 + \frac{\sigma_\ell^2}{\ell \sigma_{x'_\ell}(\sqrt{d} + \sqrt{d_{\mathrm{FFN}}})^2}. \tag{58}$$

Similarly, rewriting Equation (31) after scaling, we have

$$\left\|\frac{\partial x'}{\partial x}\right\|_2 \le 1 + \left\|\frac{\partial \mathrm{Attn}(\mathrm{LN}(x))}{\partial \mathrm{LN}(x)}\right\|_2 \left\|\frac{1}{\sqrt{\ell}}\right\|_2 \left\|\frac{\partial \mathrm{LN}(x)}{\partial x}\right\|_2. \tag{59}$$

By combining the bound (58), and inequality (59), and assuming all $\sigma$ are equal for simplicity, we obtain:

$$\left\|\frac{\partial y_L}{\partial x_1}\right\|_2 \le \prod_{l=1}^{L-1} \left(\left(1 + \frac{\sigma^2(\sqrt{d} + \sqrt{d_{\mathrm{FFN}}})^2}{\ell \sigma_{x'_\ell}}\right)\right.$$

$$\left. \times \left(1 + 2dh\left(\sqrt{s} + 2 + \frac{1}{\sqrt{s}}\right)\frac{\sigma^2}{\ell \sigma_{x_\ell}}\left(\sigma^2 d\sqrt{d_{\mathrm{head}}} + \left(1 + \sqrt{d_{\mathrm{head}}/d}\right)\right)\right)\right). \tag{60}$$

Equation (60) is a traditional product form [49] for $\sigma_{x_\ell}$. After scaling, it becomes:

$$\left\|\frac{\partial y_L}{\partial x_1}\right\|_2 \le \prod_{l=1}^{L-1} \left(1 + \frac{1}{\ell \sigma_{x_\ell}}A + \frac{1}{\ell^2 \sigma_{x_\ell}^2}B\right), \tag{61}$$

where $A$ and $B$ retain their forms from Equation (41) and Equation (42) and are treated as constants.

Regarding the upper bound on the convergence rate of $\sigma_{x_\ell}^2$, we assume $\sigma_{x_\ell}^2 = \ell^{(2-\epsilon)}$ without loss of generality. For large $L$, the product can be approximated using the properties of infinite products:

$$\prod_{\ell=1}^{L-1} \left(1 + \frac{A}{\ell^{2-\epsilon/2}} + \frac{B}{\ell^{4-\epsilon}}\right) \sim \exp\left(\sum_{\ell=1}^{L-1}\left(\frac{A}{\ell^{2-\epsilon/2}} + \frac{B}{\ell^{4-\epsilon}}\right)\right). \tag{62}$$

Then, by evaluating the sum in the exponent, we obtain:

$$\prod_{\ell=1}^{L-1} \left(1 + \frac{A}{\ell^{2-\epsilon/2}} + \frac{B}{\ell^{4-\epsilon}}\right) \sim \exp\left(A \cdot \frac{\ell^{\epsilon/2-1} - 1}{\epsilon/2 - 1} + B \cdot \frac{\ell^{\epsilon-3} - 1}{\epsilon - 3}\right). \tag{63}$$

Therefore, we establish the upper bound:

$$\left\|\frac{\partial y_L}{\partial x_1}\right\|_2 \le \Theta\left(\exp\left(A \cdot \frac{\ell^{\epsilon/2-1} - 1}{\epsilon/2 - 1} + B \cdot \frac{\ell^{\epsilon-3} - 1}{\epsilon - 3}\right)\right) = \omega(1), \tag{64}$$

where $\omega(1)$ denotes a growth strictly greater than a constant as defined before. $\qquad\square$

## C  Qwen2.5

We further evaluate the generalizability of LNS by applying it to a state-of-the-art architecture, Qwen2.5-0.5B [52]. We train the model for 6B tokens and compare LNS against the standard Pre-LN setup. Consistent with previous findings, Table 7 illustrates that LNS yields a notable reduction in perplexity—from 20.62 to 19.57—highlighting its effectiveness even on strong, modern architectures.

The consistent benefits observed across increased model scales, larger training datasets, and diverse architectures suggest that LNS is a promising technique for enhancing the training of contemporary large language models, ensuring that deeper layers contribute more effectively to learning.

Table 7: Perplexity (PPL ↓) comparison under scaled-up pre-training. For LLaMA-1B and 7B, training is scheduled for 100B tokens but is terminated early to report results. Qwen-2.5 is trained with a fixed budget of 6B tokens.

| Model | # Params | # Tokens | Pre-LN (PPL) | LNS (PPL) |
|-------|----------|----------|--------------|-----------|
| Qwen2.5-0.5B | 0.5B | 6B | 20.62 | **19.57** |

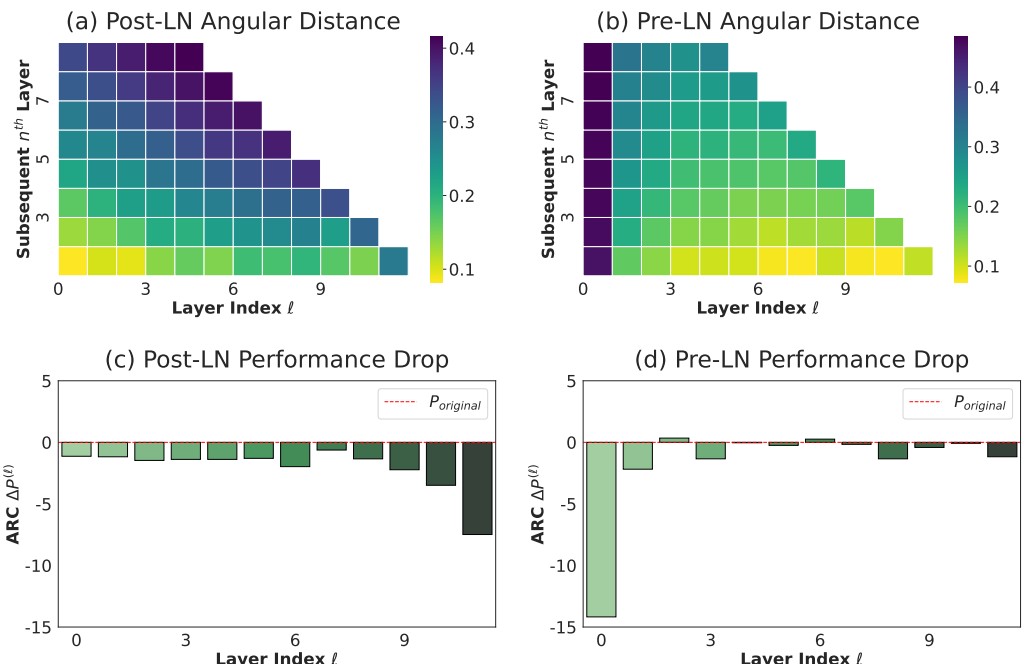

Figure 7: Results of in-house small-scale LLaMa-130M. **Angular Distance (a, b)**: Each column represents the angular distance from the initial layer $\ell$ (x-axis) and its subsequent $n^{th}$ layer (y-axis). The distance is scaled to the range [0, 1], where yellow indicates smaller distances and purple indicates larger distances. **Performance Drop (c, d)**: ARC-e performance drop of removing each single layer from LLaMa-130M.

# D    Results of In-house Small-scale LLaMa-130M

Figure 7 compares LLaMa-130M models differing only in Layer Normalization, clearly distinguishes Post-LN from Pre-LN. In Post-LN models, early layers exhibit high similarity (low angular distance, Fig. 7-a) and their removal causes minimal performance loss (Fig. 7-d), while deeper layers become more distinct and critical. Post-LN also shows larger gradients in deeper layers but severe vanishing in early layers at the start of training (Fig. 7-c). Conversely, Pre-LN LLaMa-130M demonstrates a gradual decrease in angular distance with depth, resulting in highly similar deep layers (Fig. 7-b). Removing most layers after the first in Pre-LN causes negligible performance loss (Fig. 7-d), indicating their limited contribution. These consistent findings, observed in both open-weight and in-house LLMs, lead to the conclusion that the widespread use of Pre-LN is the root cause of the ineffectiveness of deep layers in LLMs.

# E    Training Loss Curve

We report the training loss curves of Pre-LN and LayerNorm Scaling in Figure 8. While LayerNorm Scaling exhibits slightly slower convergence at the early stages of training, it consistently outperforms Pre-LN as training progresses. We attribute this to the uncontrolled accumulation of output variance in Pre-LN, which amplifies with depth and training steps, ultimately impairing the effective learning

of deeper layers. In contrast, LayerNorm Scaling mitigates this issue by scaling down the output variance in proportion to depth, thereby enabling more stable and effective training across all layers during training.

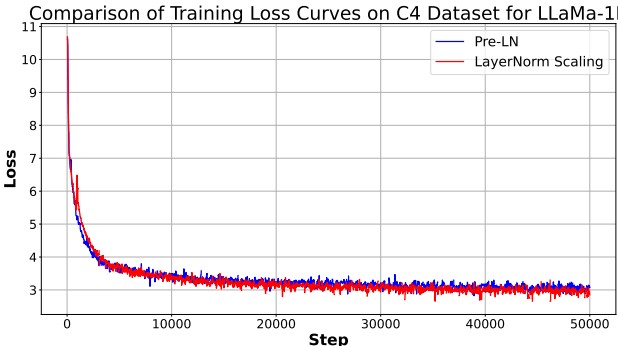

Figure 8: Training loss of LLaMA-1B with Pre-LN and LayerNorm Scaling.

## F  Variance Growth in Pre-LN Training

To analyze the impact of Pre-LN on variance propagation, we track the variance of layer outputs across different depths during training.

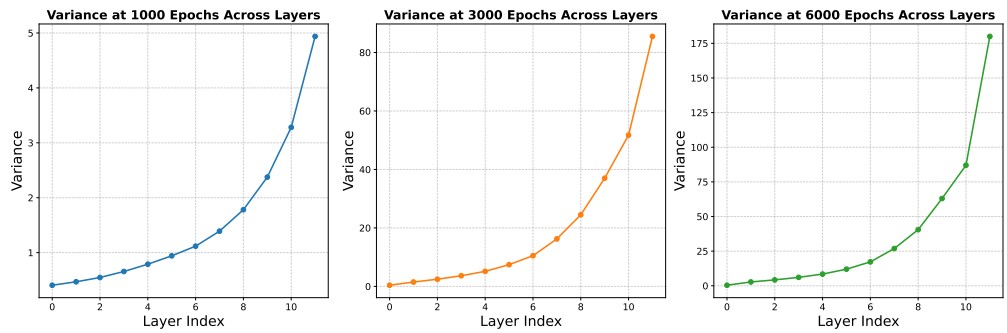

Figure 9: **Variance growth across layers in LLaMA-130M with Pre-LN.** Each subplot shows the variance at different training stages (1000, 3000, and 6000 epochs). In all cases, the variance follows an exponential growth pattern as depth increases, indicating that deeper layers experience uncontrolled variance amplification regardless of training progress.

Figure 9 illustrates the layer-wise variance in LLaMA-130M with Pre-LN at 1000, 3000, and 6000 epochs. Across all stages, variance remains low in shallow layers but grows exponentially in deeper layers, confirming that this issue persists throughout training rather than being a temporary effect. This highlights the necessity of stabilization techniques like LayerNorm Scaling to control variance and ensure effective deep-layer learning.

## G  Applicability to Vision–Language Models (Qwen 2.5-VL)

To examine whether the *Curse of Depth* also manifests in vision–language models (VLMs), we perform layer–pruning experiments on **Qwen 2.5-VL-7B** [4]. For both its vision encoder and language decoder, we prune one transformer layer at a time and directly evaluate the pruned model on the MMMU benchmark [55]. Figure 10 presents the resulting performance drops.

We observe that the **language branch** clearly suffers from the Curse of Depth, whereas the **vision branch** remains uniformly important. This suggests that the phenomenon is more pronounced in autoregressive language components of VLMs and may not directly transfer to vision encoders. A detailed modality–specific theoretical account is left to future work and community discussion.

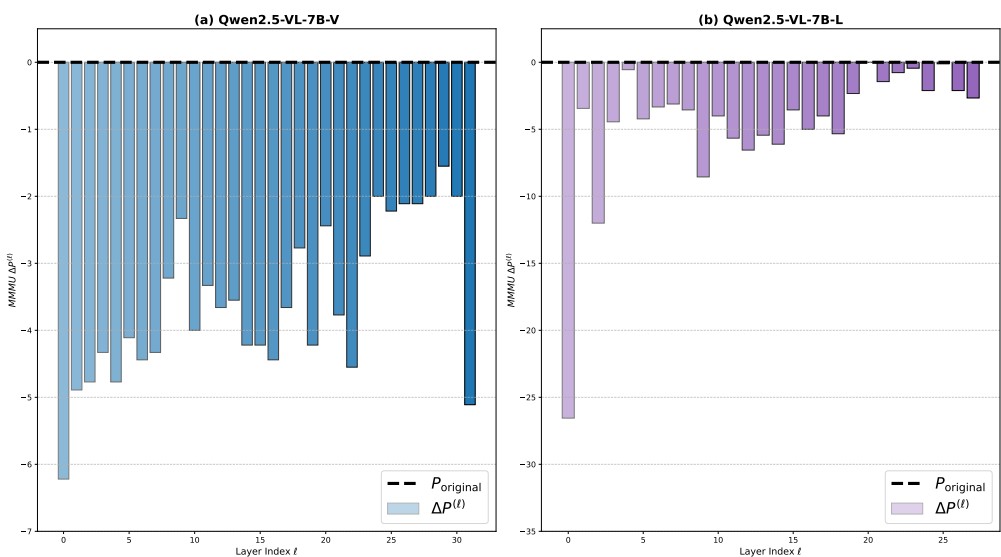

Figure 10: Performance drop of layer pruning on Qwen 2.5-VL-7B. (a) Vision branch shows relatively uniform sensitivity across layers. (b) Language branch exhibits a clear *Curse of Depth*: deeper layers contribute much less than early ones.

## H LayerNorm Scaling in Vision Transformer

To evaluate whether LayerNorm Scaling (LNS) also benefits architectures beyond language models, we conduct experiments on ViT-S for image classification. Since ViT-S includes *LayerScale* by default—which may interfere with the effect of LNS—we remove *LayerScale* from all evaluated variants to ensure a fair comparison. We then test different insertion positions of LNS. The top-1 accuracy results are summarized in Table 8. Whereas LNS in language models is typically most effective directly after normalization, in Vision Transformers, the best position is after the attention and MLP blocks. We next examine whether this performance gain correlates with better control of layer-wise variance.

Table 8: Top-1 accuracy (%) of ViT-S model with and without LNS.

| Model Variant | LNS Position | Top-1 Accuracy |
|---|---|---|
| ViT (with LayerScale) | – | 70.30 |
| ViT (w/o LayerScale) | – | 67.91 |
| ViT (w/o LayerScale) | after LayerNorm | 66.43 |
| ViT (w/o LayerScale) | after Attn/MLP | **68.75** |

Figure 11 plots the average output variance of each transformer block during training. Without LayerScale, variance in deeper layers grows rapidly—exceeding $\sim 3,000$ by 30K update steps. Applying LNS after Attn/MLP controls this growth to below $\sim 150$, confirming that LNS stabilizes the forward signal even in vision transformers.

These preliminary findings indicate that the variance-control mechanism underlying LNS generalizes to vision transformers when the scaling is applied after Attn/MLP. We leave a more detailed theoretical understanding of this behavior to future work and community discussion.

## I Post-LN Transformers

For Post-LN Transformers, we continue to adopt Assumption 1. In this setting, each layer is followed by a layer normalization (LN) step, ensuring that the variances $\sigma^2_{x_\ell}$ and $\sigma^2_{x'_\ell}$ remain fixed at 1 across

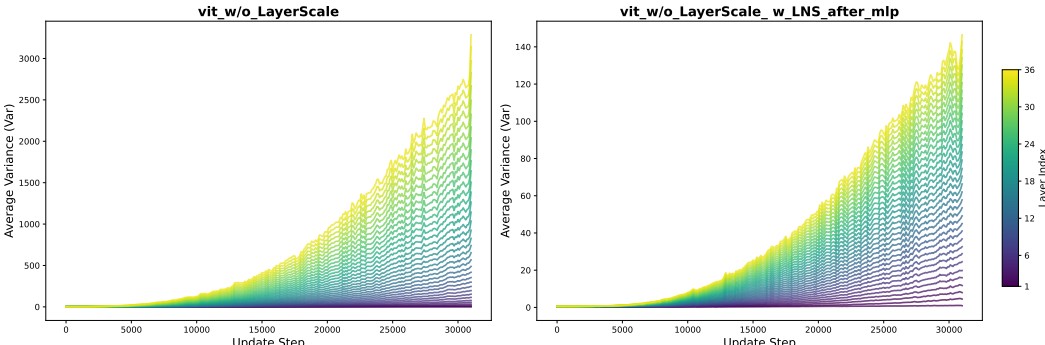

Figure 11: Layer-wise output variance of ViT-S **without** LayerScale (left) and with **LNS after Attn/MLP** (right). LNS significantly reduces the variance growth compared to the baseline.

all layers. Consequently, the norm $\left\|\frac{\partial y_\ell}{\partial x_\ell}\right\|_2$ exhibits minimal variation from one layer to the next, indicating stable gradient propagation.

Since the variance is effectively controlled by LN in Post-LN Transformers, an explicit variance-based analysis becomes less critical. Nonetheless, there remain other important aspects to investigate in deeper Post-LN architectures, such as the evolution of feature mappings and the behavior of covariance kernels over deep layers. These directions will be pursued in future work.

## J  Limitations

While this work offers a comprehensive analysis of the Curse of Depth in LLMs and proposes LayerNorm Scaling as an effective remedy, several limitations remain:

**Scope of Architectures.**  Our study primarily focuses on Transformer-based LLMs using Pre-LN. Although Pre-LN dominates modern architectures, our theoretical study does not cover models employing alternative normalization strategies (e.g., Post-LN only [13], normalization-free architectures [59]) or emerging paradigms such as mixture-of-experts or structured sparsity-based models.

**Task Coverage.**  Most empirical evaluations, including pruning and angular distance analyses, were conducted using general-purpose benchmarks like MMLU. While these tasks reflect broad model capabilities, domain-specific or long-context reasoning tasks may reveal different dynamics in deep layer contributions, which we leave for future work.

**Fine-grained Representation Quality.**  While LNS improves angular distance and performance sensitivity across layers, a deeper analysis of what types of information are represented or lost in deeper layers remains unexamined. For example, whether LNS helps preserve syntactic, semantic, or factual knowledge across depth is unclear.

## K  Broader Impact

The Curse of Depth phenomenon, identified and addressed in this work, has significant implications for the design, training, and deployment of LLMs. By revealing that deeper layers in modern Pre-LN Transformers often fail to contribute meaningfully to learning, our study prompts a reevaluation of how model capacity is allocated and utilized. This has both practical and ethical consequences.

From a computational efficiency perspective, the insights offered by this work can lead to more principled model pruning, layer reuse, or architecture design strategies that improve training and inference efficiency without compromising performance. In particular, LayerNorm Scaling enables deeper layers to be trained more effectively, maximizing the utility of each parameter and reducing unnecessary resource expenditure. This can help democratize access to powerful models by reducing

the cost of pretraining and fine-tuning, especially for institutions or communities with limited computational resources.

From a sustainability standpoint, addressing CoD has the potential to lower the environmental impact of large-scale model training by mitigating wasteful computation. With LLMs increasingly deployed in industrial-scale applications, these gains can scale into substantial reductions in energy consumption and carbon footprint.

In terms of scientific understanding, this work contributes to the growing body of research that seeks to interpret and improve the internal dynamics of deep neural networks. By identifying the gradient-preserving failure modes induced by Pre-LN at depth, we provide both a diagnosis and a remedy that could influence future research in deep optimization, normalization strategies, and interpretability.

We also caution that increasing the efficiency of LLM training and deployment through techniques like LNS may further accelerate the proliferation of powerful LLMs, raising concerns around misuse, disinformation, or labor displacement. As such, our findings should be accompanied by responsible deployment practices and continued ethical oversight in the broader AI community.

