# OpenReview forum: "The Curse of Depth in Large Language Models"
_NeurIPS.cc/2025/Conference — NeurIPS 2025 poster_

### Official Review · Reviewer_L6Vr · 2025-06-01

**Clarity:** 4
**Significance:** 4
**Originality:** 3
**Rating:** 5
**Confidence:** 3

**Summary:**

Several researchers have indicated that deep layers in LLMs are ineffective and have limited contribution to their accuracy. This paper analyzes the cause of this ineffectiveness and attributes it to pre-layer norm (i.e., placing layer norm or RMSNorm before attention and FFN). Then the paper proposes placing a simple scaling after each layer norm, and shows that it makes deeper layers more effective and increase overall accuracy of the model.

**Questions:**

- Equation 5: I didn't understand how this equation is derived. a) What is the relationship between Pre-LN(x) and LN(x)? b) why do we have an identity, I, here?
- In preliminary analysis, drop in SQuAD on BERT seems to be small, like mostly less than 1%, while drop in MMLU on other models seems to reach almost 50%. Wondering if we can verify with another task on BERT to ensure a significant variance in accuracy.
- Line 185: Even though the authors cite the experimental setup from another paper, I believe readers would appreciate if authors provide a few more details like what training set is used.
- Table 1: Please specify what evaluation data set is used to measure perplexity, and what is the context length when measuring it.
- Paper would be strong if authors could train on an architecture with MoE (e.g., Qwen3 or DeepSeekv3)
- Line 310: "Admin introduces additional parameters to control residual dependencies, stabilizing Post-LN.": Missing citation?
- Equation 19: please define n right after the equation
- I didn't understand how we derive the 3rd step in Equation 23
- if possible, it would be great if authors could do an early exit analysis similar to Figure 2 of LayerSkip paper ( https://arxiv.org/abs/2404.16710 ) to compare a model trained with LN vs. LN Scaled

**Ethical Concerns:**

["NO or VERY MINOR ethics concerns only"]

**Final Justification:**

I would like to thank the authors for their rebuttal. I have read all the reviews and rebuttals and would like to keep my score.

**Limitations:**

The author did not mention any limitations. However, I can't seem to think of a limitation.
Maybe one limitation is that auhtors did not test on a MoE model, or VLM model.

**Paper Formatting Concerns:**

- Equation 15: first parantheses is mistakenly written as subscript

**Quality:**

4

**Strengths And Weaknesses:**

Strength:
- insights backed by rigorous mathematics
- hyperparameter-free
- easy to implement
- confirmed to be stable in training (unlike other approaches proposed in literature)
- provided results training LLMs with various architectures, sizes, training budgets as well as ViTs
- Compared with a wide range of initialization / layernorm modification techniques and showed superior accuracy

Weaknesses:
- Contribution is simple

---

> ### Author Rebuttal · Authors · 2025-07-31
>
> We really thank the reviewer for recognizing the merits of our submission and valuable support! We address your remaining comments below.
>
> **Q1. Equation 5: I didn't understand how this equation is derived. a) What is the relationship between Pre-LN(x) and LN(x)? b) why do we have an identity, I, here?**
>
> **A1.** Equation (5) expresses the derivative of a residual block in a Pre-LN Transformer. In this setting, the output of each residual block can be abstractly written as: $\text{Pre-LN}(x) = x + f(\text{LN}(x))$,
> where f denotes the transformation function (e.g., attention or feed-forward sub-layer), and LN denotes the LayerNorm applied before f. Taking the derivative with respect to x, we apply the chain rule:
> $\frac{\partial\, \text{Pre-LN}(x)}{\partial x}
> = \frac{\partial}{\partial x} \left[ x + f(\text{LN}(x)) \right]
> = I + \frac{\partial f(\text{LN}(x))}{\partial \text{LN}(x)} \cdot \frac{\partial \text{LN}(x)}{\partial x}$.
>
> Our analysis finds that this derivative of deep layers in LLMs tends to approximate identity functions.
>
>
> **Q2. In preliminary analysis, drop in SQuAD on BERT seems to be small, like mostly less than 1%, while drop in MMLU on other models seems to reach almost 50%. Wondering if we can verify with another task on BERT to ensure a significant variance in accuracy.**
>
> **A2.** As you requested, we evaluate BERT on SWAG dataset, where the performance drop is more significant. We observe a similar pattern where middle to deep layers suffer more from pruning than the first few layers.
>
>
> | Layer Index   | 0      | 1      | 2      | 3      | 4      | 5      | 6      | 7      | 8      | 9      | 10     | 11     |
> |---------------|--------|--------|--------|--------|--------|--------|--------|--------|--------|--------|--------|--------|
> | Accuracy Drop | -1.74  | -1.61  | -7.51  | -8.03  | -7.41  | -8.02  | -8.12  | -10.73  | -8.34  | -6.25  | -4.38  | -2.99  |
>
>
> **Q3. Line 185: Even though the authors cite the experimental setup from another paper, I believe readers would appreciate if authors provide a few more details like what training set is used.**
> **A3.** Thanks for pointing out this! We provide the details of our training setting below.
> - Table 1: All models in Table 1 are trained from scratch on the English portion of the C4 corpus (C4-en).  We follow exactly the public training script provided in our supplementary material; the only change across runs is the choice of normalization method.  A global batch size of 512 sequences is used throughout.  For the 130M and 250M models we adopt a peak learning rate of 1 × 10⁻³, while the 350 M and 1B models use 5 × 10⁻⁴.  All other optimizer and scheduler hyper-parameters (Adam, β₁ = 0.9, β₂ = 0.95, cosine decay) remain identical across experiments.
>
> - Table 2: For the MMLU benchmark we fine-tune on its official training split.  The remaining tasks—BoolQ, ARC-easy, PIQA, HellaSwag, OBQA and Winogrande—are fine-tuned on CommonSense-170K following [1,2,3], which aggregates roughly 170k supervised examples covering these datasets.  To stabilise training, the learning rate is reduced to 1/10 of the values used during pre-training (i.e., 1 × 10⁻⁴ for < 350M models, 5 × 10⁻⁵ for larger models). Batch size is 16.
>
> - Table 3: The configuration mirrors Table 1 in every respect—same C4-en corpus, 512-sequence batches, and size-specific learning rates—but we extend the training schedule to accumulate a larger token budget.  No other hyper-parameters are altered, enabling an isolated assessment of how additional tokens (rather than different data or optimisation tweaks) influence the effectiveness of each normalization strategy.
>
> [1] Hu, Z., Wang, L., Lan, Y., Xu, W., Lim, E.P., Bing, L., Xu, X., Poria, S. and Lee, R.K.W., 2023. Llm-adapters: An adapter family for parameter-efficient fine-tuning of large language models. arXiv preprint arXiv:2304.01933.
>
> [2] Li, P., Yin, L., Gao, X. and Liu, S., 2024. Owlore: Outlier-weighed layerwise sampled low-rank projection for memory-efficient llm fine-tuning. ACL 2025.
>
> [3] Li, P., Yin, L. and Liu, S., 2024. Mix-ln: Unleashing the power of deeper layers by combining pre-ln and post-ln. ICLR 2025.
>
>
> **Q4. Table 1: Please specify what evaluation data set is used to measure perplexity, and what is the context length when measuring it.**
>
> **A4.** We follow the evaluation protocol introduced in Galore [1]. Perplexity is computed on the C4 validation split (the “en” portion of the Colossal-Clean-Crawled-Corpus). Each document is chunked into 256-token contexts without overlap, and perplexity is averaged across all resulting sequences.
>
> [1] Zhao, J., Zhang, Z., Chen, B., Wang, Z., Anandkumar, A. and Tian, Y., 2024. Galore: Memory-efficient llm training by gradient low-rank projection. ICML 2024.
>
> **Q5. Paper would be strong if authors could train on an architecture with MoE (e.g., Qwen3 or DeepSeekv3)**
>
> **A5.** We fully agree that applying LayerNorm Scaling to a modern MoE architecture would further validate the generality of our approach. Unfortunately, training a competitive MoE model from scratch exceeds the compute budget and time available during the short rebuttal window. We plan to explore this promising extension after the rebuttal period, when more resources and time are available, and we will include the results in our next version.
>
>
> **Q6. Line 310: "Admin introduces additional parameters to control residual dependencies, stabilizing Post-LN.": Missing citation?**
>
> **A6.** Thanks for your detailed comment! The Admin approach was proposed by Liu et al [1]. We will add it in our revision.
>
> [1] Liu, L., Liu, X., Gao, J., Chen, W. and Han, J., 2020. Understanding the difficulty of training transformers. arXiv preprint arXiv:2004.08249.
>
>
> **Q7. Equation 19: please define n right after the equation**
>
> **A7.** We appreciate your careful observation! In Equation (19), both 𝑛 and 𝑁 refer to the number of attention heads. We acknowledge the inconsistency and will revise the notation in the next version to improve clarity and maintain consistency.
>
>
> **Q8. I didn't understand how we derive the 3rd step in Equation 23**
>
> **A8.** Thank you for the question. The third step in Equation (23) uses asymptotic notation. Specifically, the expression $1 + \frac{\sigma_W}{\sigma_{x_\ell}} + \frac{\sigma_W^2}{\sigma_{x_\ell}^2}$ is approximated by $\Theta\left(1 + \frac{1}{\sigma_{x_\ell}}\right)$
>
> Here, the notation $\Theta(\cdot)$ denotes the dominant growth rate of the variance term defined in Lemma 1, with respect to $\sigma_{x_\ell}$. Both $\frac{\sigma_W}{\sigma_{x_\ell}}$ and $\frac{\sigma_W^2}{\sigma_{x_\ell}^2}$ decrease as $\sigma_{x_\ell}$ becomes large. In the final approximation, we retain only the dominant term and omit higher-order terms for simplicity, resulting in the asymptotic form $\Theta\left(1 + \frac{1}{\sigma_{x_\ell}}\right)$.
>
>
>
>
>
> **Q9. if possible, it would be great if authors could do an early exit analysis similar to Figure 2 of LayerSkip paper ( https://arxiv.org/abs/2404.16710 ) to compare a model trained with LN vs. LN Scaled**
>
> **A9.** We thank the reviewer for the thoughtful suggestion to evaluate our method in combination with LayerSkip. We agree that this is a very intriguing direction. However, applying LayerSkip effectively requires large-scale pretraining or continued pretraining (>=52B), which is unfortunately beyond our computational budget within the limited rebuttal period. We fully acknowledge the potential of combining these two approaches and plan to explore this promising direction in future work.

---

### Official Review · Reviewer_tprK · 2025-07-01

**Clarity:** 2
**Significance:** 3
**Originality:** 3
**Rating:** 4
**Confidence:** 4

**Summary:**

The paper identifies and analyzes a phenomenon the authors call the Curse of Depth(CoD) in Pre-LayerNorm (Pre-LN) transformers. Empirically, the top-most blocks of several mainstream LLMs (up to 40 layers in 13 B models) can be pruned with negligible loss, indicating that these layers contribute little to representation learning. The authors attribute CoD to an exploding activation variance along residual paths, which forces the layerwise Jacobian toward the identity and suppresses gradient flow. They prove upper- and lower-bounds on this variance growth, then propose a one-line fix LayerNorm Scaling (LNS), i.e., dividing each LayerNorm output by $1/\sqrt{\ell}$. LNS theoretically limits variance to polynomial growth, restores gradient magnitudes, and empirically improves perplexity, downstream accuracy, and the functional importance of deep layers in 130 M–1 B models.

**Questions:**

1. **Down-stream Metrics on 7B**

    In Section 5.3, you report perplexity improvements for a 7B (32-layer) model, but no MMLU, ARC-e, or other accuracy numbers. Did you run those tests and, if so, could you share the results? If not, please clarify why only PPL was measured at that scale.

2. **Ultra-deep Applicability (≥ 80 layers)**

    The deepest setting evaluated for LayerNorm Scaling (LNS) is 32 layers. Have you analyzed variance curves, pruning sensitivity, and task accuracy on checkpoints in the 80–120-layer range (e.g., LLaMA-2-70 B)? Even a post-hoc fine-tune would help confirm whether the proposed fix remains effective in the truly “deep” regime.

3. **Tail-layer Importance**

    The upward “tail” in pruning loss suggests that the last few blocks may perform specialized functions (e.g., output-projection alignment). Do you foresee that a uniform $1/\sqrt{\ell}$ schedule could dampen such specialization, and have you measured any task where the very last layer dominates after LNS is applied?

4. **Resource Foot-print Quantification**

    One motivation for LNS is to “avoid wasting compute on dead layers.” Could you provide wall-clock time, GPU-memory, and energy usage comparisons between Pre-LN and LNS models at 130 M, 400 M, and 1 B scales?

5. **Exponent Search and Adaptive Scaling**

    Your experiments use a fixed $1/\sqrt{\ell}$ factor. Did you explore alternative exponents $1/l^\beta (\beta\in[0.3, 0.7])$ or a run-time variance-adaptive schedule? Please report any trends in stability and accuracy.

6. **RMSNorm ablation and cross-norm mixes**

    All results keep RMSNorm active. Could you (i) disable RMSNorm and apply LayerNorm + LNS only; (ii) combine LNS with DeepNorm or residual α-scaling; and (iii) report variance and accuracy in each setting? This would isolate the true contribution of LNS versus RMSNorm.

**Ethical Concerns:**

["NO or VERY MINOR ethics concerns only"]

**Final Justification:**

1. **Clear problem identification**
   * Shows that upper transformer blocks add almost no value.
   * Targets the industry-standard *Pre-LayerNorm + Residual* architecture, making the diagnosis immediately relevant.

2. **Tight theory-experiment loop**
   * Theorizes exponential activation-variance growth and Jacobian collapse.
   * Logs training stats to confirm the variance explosion and gradient/pruning patterns.
   * Introduces LayerNorm Scaling ($1/\sqrt{\ell}$) and re-applies the same analysis to show variance drops to polynomial growth and deep layers regain utility.

3. **Minimal, broadly applicable fix**
   * One-line code change, no extra parameters.
   * Consistently lowers perplexity and boosts accuracy on 130 M–1 B models, with potential to extend to larger checkpoints.

**Limitations:**

yes

**Paper Formatting Concerns:**

1. Several equations in the appendix are mis-numbered or not rendered correctly. For example, references to “(28)” and “(38)” point to expressions whose numbers do not appear; in a few places the equation label is replaced by a blank square.
2. Checklist item 16 is missing from the submission.

**Quality:**

3

**Strengths And Weaknesses:**

### Strengths
1. **Clear problem identification**
   * Shows that upper transformer blocks add almost no value.
   * Targets the industry-standard *Pre-LayerNorm + Residual* architecture, making the diagnosis immediately relevant.

2. **Tight theory-experiment loop**
   * Theorizes exponential activation-variance growth and Jacobian collapse.
   * Logs training stats to confirm the variance explosion and gradient/pruning patterns.
   * Introduces LayerNorm Scaling ($1/\sqrt{\ell}$) and re-applies the same analysis to show variance drops to polynomial growth and deep layers regain utility.

3. **Minimal, broadly applicable fix**
   * One-line code change, no extra parameters.
   * Consistently lowers perplexity and boosts accuracy on 130 M–1 B models, with potential to extend to larger checkpoints.

---

### Weaknesses
1. **Limited experimental scale**
   * Remedy is tested only up to 1 B (24 layers). Behavior on 70 B–110 B models (80–120 layers) is unverified; the $1/\sqrt{\ell}$ schedule may need retuning.

2. **Incomplete factor isolation**
   * All runs keep RMSNorm, so gains from the new scaling vs. RMSNorm stability aren’t disentangled.
   * No comparisons with alternatives like DeepNorm or residual α-scaling.
   * Mean-field assumptions (Gaussian, independent channels) are not empirically checked late in training.

3. **Practical trade-offs and reproducibility gaps**
   * No data on speed, memory, or energy—so efficiency benefits are unclear.
   * Key hyper-parameters (batch size, grad clipping, weight decay) are under-documented.
   * Evaluation omits code generation, dialogue, and reasoning tasks where deep context might matter most.

---

> ### Author Rebuttal · Authors · 2025-07-31
>
> Thanks for your time and your constructive comments!
>
> **Q1. Limited experimental scale
> Behavior on 70 B–110 B models (80–120 layers) is unverified; the schedule may need retuning. Even a post-hoc fine-tune would help confirm whether the proposed fix remains effective in the truly “deep” regime.**
>
> **A1:** We appreciate the reviewer’s comments. Unfortunately, training models of this size remains prohibitively expensive for academic researchers without access to industry-scale infrastructure. Moreover, since large pre-trained models have been trained with Pre-LN with massive data, it is difficult to directly apply LNS to them in a post-hoc fine-tuning manner.
>
> Nevertheless, we conducted extra scaling experiments to provide **strong evidence for the scalability of our approach** through a scaling training of 60M, 150M, 300M, 1B, and 7B parameters under a fixed 20B-token budget. We use the same scaling factor i.e., $1/\sqrt{l}$, for all models. LNS consistently and substantially outperforms the standard Pre-LN baseline across all model sizes.  These results highlight the robustness and scalability of our method, suggesting that its benefits are likely to persist—and potentially amplify—at even larger model scales.
>
> |  Model Size        | # Tokens | Pre-LN  | LNS  |
> |---------------|----------|--------------|-----------|
> | 60M      | 20B       |   3.67    | **3.56** |
> | 150M     | 20B       |    3.44   | **3.24** |
> | 300M     | 20B       |    3.29   | **3.14** |
> | 1B       | 20B       |   3.15    | **2.85** |
> | 7B       | 20B       |   2.69    | **2.50** |
>
> **Q2. Incomplete factor isolation.
> All runs keep RMSNorm, so gains from the new scaling vs. RMSNorm stability aren’t disentangled...**
>
> **A2:** We want to clarify that the gain is indeed achieved by our LayerNorm Scaling **not by RMSNorm**, since RMSNorm is used for all approaches. As you requested, we further disable RMSNorm and apply LayerNorm with LNS. We can see that the standard LayerNorm+LNS also outperforms LayerNorm only, isolating the true contribution of LNS versus RMSNorm. LNS effectively scales down the output variance of LayerNorm from 67 to 11.
>
>
> | Model        | Method         | Perplexity | Variance (layer12) |
> |--------------|----------------|------------|------------|
> | LLaMA-130M   | Pre-LN (RMS)         | 26.73    |   71.00
> | LLaMA-130M   | Pre-LN (RMS) + LNS   | **25.76**    |    12.13
> | LLaMA-130M   | Pre-LN (LayerNorm)   | 26.71   | 67.00
> | LLaMA-130M   | Pre-LN (LayerNorm) + LNS   | **25.74**    |11.19
>
> **Combining LNS with Other Methods.** DeepNorm inherently avoids variance explosion since its outputs are always normalized. Consequently, applying LNS on top of DeepNorm causes training divergence. In contrast, α-scaling does not address variance explosion, whereas LNS effectively mitigates it, resulting in improved performance.
>
>
> | Model        | Method         | Perplexity | Variance (layer12) |
> |--------------|----------------|------------|------------|
> | LLaMA-130M   | DeepNorm    | 27.17 |  1.00
> | LLaMA-130M   | DeepNorm + LNS   | 1335.92 |    0.09
> | LLaMA-130M   | Pre-LN (RMS)   | 26.73    |   71.00
> | LLaMA-130M   | Pre-LN (RMS) + α-scaling     | 26.69   | 71.50
> | LLaMA-130M   | Pre-LN (RMS) + α-scaling + LNS | **25.73**   |11.44
>
> **Q3: Mean-field assumptions (Gaussian, independent channels) are not empirically checked late in training.**
>
> **A3:** We acknowledge that the proofs of Lemma 1 and Theorem 1 are based on standard idealized assumptions (Gaussian, independent channels). This setting is widely used in statistical machine learning analysis, including Takase et al. (2023) and Xiong et al. (2020) that we rely on. It allows us to precisely analyze the effect of variance scaling. It captures key behaviors relevant to gradient propagation, and extending to more general settings is an important direction for future work.
>
>
> **Q4. No data on speed, memory, or energy—so efficiency benefits are unclear. One motivation for LNS is to “avoid wasting compute on dead layers.”**
>
> **A4:** By "avoiding wasting compute on dead layers," we **do not** mean that our primary goal is to reduce computational cost—a topic extensively explored in prior work on layer pruning [1,2,3]. Instead, our objective is to ensure that deeper layers contribute more effectively to learning, so that the substantial resources used for training these layers are not wasted, ultimately leading to stronger and more capable models. This is supported by our layerwise similarity analysis in our paper, which shows that LNS improves the feature diversity across layers, as well as the stronger evaluation performance achieved by LNS.
>
> Apart from this, LNS is a simple scaling operation applied after LayerNorm, introducing almost no additional computational overhead compared to standard LayerNorm.
>
> [1] Men, X., Xu, M., Zhang, Q., Wang, B., Lin, H., Lu, Y., Han, X. and Chen, W., 2024. Shortgpt: Layers in large language models are more redundant than you expect. arXiv preprint arXiv:2403.03853.
>
> [2] Gromov, Andrey, Kushal Tirumala, Hassan Shapourian, Paolo Glorioso, and Daniel A. Roberts. "The unreasonable ineffectiveness of the deeper layers." arXiv preprint arXiv:2403.17887 (2024).
>
> [3] Yang, Y., Cao, Z. and Zhao, H., 2024. Laco: Large language model pruning via layer collapse. arXiv preprint arXiv:2402.11187.
>
> **Q5. Key hyper-parameters (batch size, grad clipping, weight decay) are under-documented.**
>
> **A5.** We closely follow the training configurations used in prior studies [1,2], ensuring a fair and consistent comparison across all baselines. A detailed description of the hyperparameters has been added in the revised version.
>
> [1] Zhao, Jiawei, Zhenyu Zhang, Beidi Chen, Zhangyang Wang, Anima Anandkumar, and Yuandong Tian. "Galore: Memory-efficient llm training by gradient low-rank projection." ICLM 2024.
>
> [2] Li, P., Yin, L. and Liu, S., 2024. Mix-ln: Unleashing the power of deeper layers by combining pre-ln and post-ln. ICLR 2025.
>
> **Q6. Evaluation omits code generation, dialogue, and reasoning tasks where deep context might matter most.**
>
> **A6.** Thank you for your insightful question. Due to our limited computational resources, we pre-trained our models on a relatively small number of tokens, which is insufficient to achieve competitive performance on formal reasoning tasks such as code generation or mathematical problem solving.
>
> Nevertheless, it is well-established that stronger base models tend to exhibit better reasoning capabilities. Moreover, model depth plays a crucial role in reasoning, as highlighted by the Physics of Language Models paper (https://arxiv.org/pdf/2407.20311). Our proposed LNS improves the effectiveness of deeper layers, thereby enhancing the model’s capacity for complex reasoning. We are confident that our approach can lead to improved performance on formal reasoning benchmarks when applied to larger-scale models.
>
> **Q7. Down-stream Metrics on 7B
> In Section 5.3, you report perplexity improvements for a 7B (32-layer) model, but no MMLU, ARC-e, or other accuracy numbers.**
>
> **A7:** As you requested, we further evaluate the zero-shot performance of our models on various reasoning tasks. The results show that the improvements in PPL effectively transfer to downstream metrics. Our method delivers a notable average improvement of 2.1%, demonstrating its practical benefit beyond pre-training loss/PPL.
>
> | Model/Metric   | ARC-easy | COPA | HellaSwag | PIQA | Winogrande | SciQ | **Average** |
> |----------------|---------|------|-----------|------|------------|------|------------|
> | **Pre-LN (7B)**| 59.8    | 71.0 | 55.6      | 71.7 | 53.9       | 84.8 | **66.1** |
> | **LNS (7B)**   | 61.4    | 76.0 | 57.1      | 74.8 | 55.2       | 84.9 | **68.2** |
>
> **Q6. Tail-layer Importance
>  Do you foresee that a uniform schedule could dampen such specialization, and have you measured any task where the very last layer dominates after LNS is applied?**
>
> **A6.** Very good questions! We agree with you that the last few blocks perform specialized functions that are crucial for the performance. However, our uniform scaling is designed primarily to enhance the contribution of these layers. As a result, these final blocks actually become more crucial, rather than being suppressed. To verify this, we conducted an additional experiment where we excluded the last three layers from scaling. The results (shown below) indicate that this “no-scaling” variant performs worse than uniform scaling with $1/\sqrt(l)$, although it still provides some improvement over the standard Pre-LN baseline.
>
>
> | Model        | Method           | Tail Layer Scaling  | Perplexity |
> |--------------|------------------|---------------------|------------|
> | LLaMA-130M   | Pre-LN           | ✗                   | 26.95      |
> | LLaMA-130M   | Pre-LN + LNS     | ✗ (last 3 layers)   | 26.61      |
> | LLaMA-130M   | Pre-LN + LNS     | ✓ (all layers)      | **25.76**  |
>
> Regarding your second question, our preliminary results show that the middle and deep layers indeed become more important for reasoning-intensive tasks such as GSM8K. However, we do not observe that the last few layers can dominate over the first few layers, likely because the model still uses a Pre-LN architecture, even though LNS enhances the effectiveness of deeper layers.
>
> **Q7. Exploring alternative exponents $1/l^\beta, \beta \in[0.3, 0.7]$ or a run-time variance-adaptive schedule?**
>
> **A7.** As you requested, we run experiments with $\beta=0.3$ and $\beta=0.7$. The results below show that $\beta=0.5$ (our setting) achieves the best performance.
>
> | Model        | Method         | Exponent        | Perplexity |
> |--------------|----------------|-----------------|------------|
> | LLaMA-250M   | Pre-LN         | -               | 21.92      |
> | LLaMA-250M   | Pre-LN + LNS   | 0.3             | 20.51      |
> | LLaMA-250M   | Pre-LN + LNS   | 0.5 (ours)    | **20.35**  |
> | LLaMA-250M   | Pre-LN + LNS   | 0.7             | 20.99

---

> > ### Comment · Reviewer_tprK · 2025-08-07
> >
> > First of all, I would like to **thank the authors for addressing most of the questions** to the best of their ability.
> >
> > However, in my previous review, I had asked **why the evaluation was limited to only perplexity (PPL)**. Specifically, I inquired about the reason for not including additional tasks commonly used to assess LLM performance(i.e. MMLU, commonsense reasoning, etc.). Unfortunately, I did not receive a response regarding this point.
> >
> > I also remain unconvinced about whether the performance **differences across various models presented in the Pre-LN vs. LNS comparison are truly significant**. For instance, in the case of the 7B model, the gap in **PPL is only 0.19**, and it is unclear whether such a difference would translate into meaningful gains in other tasks.
> >
> > In addition, I had requested further experiments using alternative scaling factors **in place of the square root in** $1 / \sqrt{l} $ , but this was not addressed in the rebuttal.
> >
> > Nonetheless, **I appreciate the authors’ efforts to prepare a detailed response** within the limited time frame.

---

> ### Author Response · Authors · 2025-08-07
> **Thank you for your response**
>
> We sincerely thank Reviewer **tprK** for the thoughtful feedback.
>
> It appears that some of our previous responses may have been **overlooked**. We would like to clarify that we have already addressed **all** the points raised and provided the requested experiments. It might be because our repeated bullet numbers caused your misunderstanding. For your convenience, we summarize them again below:
>
> ---
>
> ### 1. Downstream Task Performance
>
> As shown in our previous response (**A7**), we conducted downstream evaluations on a suite of commonsense reasoning tasks. Our method demonstrates a consistent performance boost, achieving an average improvement of **2.1%** over the Pre-LN baseline:
>
> | Model/Metric   | ARC-easy | COPA | HellaSwag | PIQA | Winogrande | SciQ | **Average** |
> |----------------|----------|------|-----------|------|-------------|------|------------|
> | **Pre-LN (7B)**| 59.8     | 71.0 | 55.6      | 71.7 | 53.9        | 84.8 | **66.1**    |
> | **LNS (7B)**   | 61.4     | 76.0 | 57.1      | 74.8 | 55.2        | 84.9 | **68.2**    |
>
> ---
>
> ### 2. PPL is only 0.19
>
> In response to your concern about the seemingly small gain (0.19), we clarify that this value refers to the **training loss**, **not** perplexity (PPL). This **0.19 loss reduction** translates into a **2.55 PPL improvement**, which we consider substantial.
>
> ---
>
> ### 3. Alternative Scaling Factors
>
> As shown in our last answer, we conducted experiments with different scaling exponents to validate robustness. Our default setting (exponent = 0.5) consistently achieves the best performance:
>
> | Model        | Method         | Exponent       | Perplexity |
> |--------------|----------------|----------------|------------|
> | LLaMA-250M   | Pre-LN         | –              | 21.92      |
> | LLaMA-250M   | Pre-LN + LNS   | 0.3            | 20.51      |
> | LLaMA-250M   | Pre-LN + LNS   | **0.5 (ours)** | **20.35**  |
> | LLaMA-250M   | Pre-LN + LNS   | 0.7            | 20.99      |
>
> ---
>
> We hope this clears up any misunderstandings, and we remain available to further clarify if needed. Thank you again for your time and consideration.

---

### Official Review · Reviewer_skvN · 2025-07-01

**Clarity:** 2
**Significance:** 4
**Originality:** 3
**Rating:** 5
**Confidence:** 3

**Summary:**

This paper explores the "curse of depth," which is that later layers seem to contribute less than earlier layers. This is demonstrated by experiment, and a theoretical analysis shows that gradients have a constant upper bound as the depth of the network increases. The authors propose an extremely simple modification that scales the contribution of layer l by 1/sqrt(l). They repeat the theoretical analysis and show that the gradient grows linearly in the depth, and they repeat the experiments showing that the layers contribute more equally. Finally, they show through extensive experiments that their modification improves language model performance.

**Questions:**

- I found the theoretical argument around Lemma 1 and Theorem 1 difficult to follow. If the goal is to prevent the later layers from becoming identity-like -- that is, dominated by the residual connections -- it seems counterintuitive that the solution would be to scale down the part of each layer that is not the residual connection. Could you provide some explanation to resolve this paradox?

- Since each layer norm has a scaling factor built into it (usually called $\gamma$), why couldn't it learn the $\sqrt{\ell}$ factor automatically?

- In Figure 2b, what’s the reason for the spike around layer 27-28?

- In Eq (23), $\sigma_W^2$ should be $\sigma_W^4$ and $\sigma_W$ should be $\Sigma_W^2$ (right?)

The proof of Lemma 3, seems, to me, to skip a lot of steps.
- How do you get from (32) and (33) to (34)? It would help to write out the Jacobian of LN.
- How do you get equation (37)? The only justification is a citation to an entire textbook [44].

**Ethical Concerns:**

["NO or VERY MINOR ethics concerns only"]

**Final Justification:**

Thank your for your responses. I don't have anything to add and will keep my score the same.

**Limitations:**

The limitations section is relegated to the appendix but is adequate.

**Quality:**

4

**Strengths And Weaknesses:**

# Strengths

Both the experimental and theoretical results are convincing, and the LNS modification is so simple that the paper makes a very compelling case that it should be adopted.

# Weaknesses

I found the theoretical argument around Lemma 1 and Theorem 1 difficult to follow. If the goal is to prevent the later layers from becoming identity-like -- that is, dominated by the residual connections -- it seems counterintuitive that the solution would be to scale down the part of each layer that is not the residual connection. Could you provide some explanation to resolve this paradox?

# Minor comments

- line 243: the reference to Figure 3 is not right. I couldn’t find what the right referent should be.

---

> ### Author Rebuttal · Authors · 2025-07-31
>
> We sincerely thank the reviewer for the positive evaluation. We are pleased that they find our experimental and theoretical results convincing and consider LNS compelling for adoption. We address your remaining comments below.
>
>
> **Q1. I found the theoretical argument around Lemma 1 and Theorem 1 difficult to follow. If the goal is to prevent the later layers from becoming identity-like -- that is, dominated by the residual connections -- it seems counterintuitive that the solution would be to scale down the part of each layer that is not the residual connection. Could you provide some explanation to resolve this paradox?**
>
> **A1.** Thanks for your great question. According to our theoretical analysis, the Jacobian matrix of a full Transformer layer—including both residual and non-residual components—tends to approximate the identity matrix. This means the entire layer contributes minimally during the backward pass. While scaling down non-residual part will amplify the contribution of the residual part, the Jacobian matrix of the whole layer becomes non-identity, therefore, making more meaningful contributions to training.
>
>
> **Q2. line 243: the reference to Figure 3 is not right. I couldn’t find what the right referent should be.**
>
> **A2.** Figure 3 is the illustration of Pre-LN and LayerNorm Scaling. Figure 3(a) is the Pre-LN and Figure 3(b) is the LayerNorm Scaling. We will revise the reference in the text to clarify this in the final version.
>
>
>
> **Q3. Since each layer norm has a scaling factor built into it (usually called ), why couldn't it learn the  factor automatically?**
>
> **A3.** Thank you for the insightful question. We appreciate your attention to the role of the learnable parameter $\gamma$ in LayerNorm. Our analysis reveals that $\gamma$ increases the output variance by amplifying the magnitude of outliers in LLM activations—a phenomenon also observed in Figure 1 of [1]. In other words, $\gamma$ exacerbates the "curse of depth" in deep networks.
>
> Our plausible explanation is that Pre-LN together with skip connection is designed to improve the stability of deep Transformers training. However, it tends to take the short-cut and driving the network toward an extreme behavior, i.e., causing deeper layers to approximate identity functions. While this enable training very deep Transformers (i.e., allowing many layers to be stacked without vanishing gradients), it comes at the cost of reduced expressiveness and diminished learning capacity in deeper layers.
> To mitigate this issue, we propose explicitly scaling down the output variance, enabling more effective learning in deeper layers.
>
>
> **Q4. In Figure 2b, what’s the reason for the spike around layer 27-28?**
>
> **A4.** This is due to the distinct roles played by different layers in the network. The last few layers are responsible for transforming high-level representations into final outputs (e.g., classification logits). As a result, any error or disruption in these layers directly affects the model’s performance, making it less tolerant to their removal compared to the middle layers. In contrast, the middle-to-deep layers often contain considerable redundancy, allowing the representational loss from their removal to be partially compensated by adjacent layers.
>
>
>
>
> **Q5. In Eq (23), $\sigma_W^2$ should be $\sigma_W^4$ and $\sigma_W$ should be $\sigma_W^2$ (right?)**
>
> **A5.** Thank you for your careful observation. Equation (23) characterizes the relationship between $x_\ell$ and $x^\prime_\ell$, rather than between $x_\ell$ and $x_{\ell-1}$. In this case, we are considering the output of a single attention layer. Therefore, the variance and standard deviation terms remain as $\sigma_W^2$ and $\sigma_W$, respectively, rather than $\sigma_W^4$ and $\sigma_W^2$.
>
>
> **Q6. How do you get from (32) and (33) to (34)? It would help to write out the Jacobian of LN.**
>
> **A6**. Thank you for the comment. This part of the derivation is a simplified version of the analysis presented in [2], specifically Section 3.
>
> Specifically, from Equation (33), we know that:
> $$
> \left\lVert \frac{\partial \mathrm{FFN}(\mathrm{LN}(x'))}{\partial \mathrm{LN}(x')} \right\rVert_{2} \leq \sigma_1 \sigma_2 \left( \sqrt{d} + \sqrt{d_{\mathrm{ffn}}} \right)^2,
> $$
>
> where the right-hand side satisfies the identity $\sigma_1 \sigma_2 (\sqrt{d} + \sqrt{d_{\mathrm{ffn}}})^2 = \lVert W_1 \rVert_2 \lVert W_2 \rVert_2$.
>
> Next, regarding the LN part, the Jacobian matrix of LN can be written as:
> $$
> \frac{\partial \mathrm{LN}(x')}{\partial x'}
> = \frac{\sqrt{d}}{\lVert x' \rVert_2} \left( I - \frac{x' x'^\top}{\lVert x'\lVert_{2}^2} \right)
> = \frac{\sqrt{d}}{\sigma_{x'} \sqrt{d}} \left( I - \frac{x' x'^\top}{\sigma_{x'}^2 d} \right)
> = \frac{1}{\sigma_{x'}} \left( I - \frac{z z^\top}{d} \right).
> $$
>
> The second equation uses $\lVert x'\rVert_2 = \sigma_{x'} \sqrt{d}$, which can be obtained based on Assumption 1. The last equation is derived from the well-known formula of $z = (x' - \mu_{x'}) / \sigma_{x'}$, which converts a normal distribution, $x'$, to the standard normal distribution $z$, where $\mu_{x'} = 0$ in Assumption 1.
>
>
> We consider the variance ($\mathrm{var}$) of each element in the matrix $zz^\top$. Since $z_i z_i$ follows $\chi^2$ with 1 degree of freedom, and $z_i z_j$ ($i \neq j$) is the multiplication of two independent values following the standard normal distribution, the variances are as follows:
> \begin{align}
> \mathrm{var}(z_i z_j) =
> \begin{cases}
> 1 & \text{if } i \neq j \\\\
> 2 & \text{otherwise}
> \end{cases}
> \end{align}
>
> It indicates that $\frac{zz^\top}{d} \sim 0$ in LLMs due to $d \gg 1$. Therefore, the spectral norm of the Jacobian matrix of LN can be written as:
> \begin{align}
> \left\lVert \frac{\partial \mathrm{LN}(x')}{\partial x'} \right\rVert_{2}
> = \frac{1}{\sigma_{x'}}, \quad
> \text{where} \quad
> \frac{\partial \mathrm{LN}(x')}{\partial x'} = \frac{1}{\sigma_{x'}} I.
> \end{align}
>
> Finally, combining the above equations can be rewritten as:
> $$
> \left\lVert \frac{\partial y}{\partial x'} \right\rVert_{2}
> \leq 1 + \frac{\sigma_1 \sigma_2}{\sigma_{x'}} (\sqrt{d} + \sqrt{d_{\mathrm{ffn}}})^2.
> $$
>
>
> According to the Assumption 1 and discussion in the paper, the standard deviations, $\sigma_1$ and $\sigma_2$, of $W_1$ and $W_2$, respectively, should be sufficiently small, and the standard deviation, $\sigma_{x'}$, of the shortcut $x'$, should satisfy $\sigma_1 \sigma_2 \ll \sigma_{x'}$ in order to keep the upper bound small. Here the right-hand side satisfies the identity $\sigma_1 \sigma_2 (\sqrt{d} + \sqrt{d_{\mathrm{ffn}}})^2 = \lVert W_1\rVert_{2} \lVert W_2 \lVert_{2}$.
>
>
> **Q7. How do you get equation (37)? The only justification is a citation to an entire textbook [44].**
>
> **A7.** Equation (37) is also derived from the analysis in [2], specifically from Appendix Equation (53). The analysis employs $\lVert W\rVert_2 \sim \sigma (\sqrt{d_{in}} + \sqrt{d_{out}})$ from Vershynin (2018), High-Dimensional Probability. The statement is located in Chapter 4, Section 4.4, around Theorem 4.4.5 and Exercise 4.4.7 the book. We additionally cite Vershynin 2018 here for completeness.
>
> Specifically,
>
>
> $$
> \begin{aligned}
> \left\lVert J_{kj} \right\rVert_2
> &= \left\lVert \sum_{l=1}^{L} \left( \frac{\partial A_{kl}}{\partial x_j} v_l^\top \right) + A_{kj} W_V \right\rVert_2 \\\\
> &\leq \sum_{l=1}^{L} \left\lVert \left( \frac{\partial A_{kl}}{\partial x_j} v_l^\top \right) \right\rVert_2 + \lVert A_{kj} W_V \rVert_2 \\\\
> &\leq \frac{1}{\sqrt{L}} \left( \delta_{kj} + 1 + \frac{2}{\sqrt{L}} \right) \sigma^3 \sqrt{d^3 d_{\mathrm{head}}} + \frac{\sigma}{L} \left( \sqrt{d} + \sqrt{d_{\mathrm{head}}} \right)
> \end{aligned}
> $$
>
> $$
> \begin{aligned}
> \left\lVert J^Z \right\rVert_2
> &= \left\lVert \sum_{k=1}^{L} \sum_{i=1}^{h} J^i_{kj} W_i \right\rVert_2 \\\\
> &\leq \sum_{k=1}^{L} \sum_{i=1}^{h} \lVert J^i_{kj} \rVert_2 \lVert W_i \rVert_2 \\\\
> &\sim h \left( \frac{1}{\sqrt{L}} \left(1 + L + \frac{2L}{\sqrt{L}} \right) \sigma^3 \sqrt{d^3 d_{\mathrm{head}}} + \frac{L\sigma}{L} \left( \sqrt{d} + \sqrt{d_{\mathrm{head}}} \right) \right) \\\\
> &= h \left( \left( \sqrt{L} + 2 + \frac{1}{\sqrt{L}} \right) \sigma^3 \sqrt{d^3 d_{\mathrm{head}}} + \sigma \left( \sqrt{d} + \sqrt{d_{\mathrm{head}}} \right) \right)
> \end{aligned}
> $$
>
> Here $\lVert J^Z \rVert_2$ is the norm of Jacobian matrix and $\lVert J^i_{kj} \rVert_2$ is a specific term of $\lVert J^Z \rVert_2$. Applying $\lVert J_{kj} \rVert_2$ to above equation we can get Equation (37) in our paper.
>
> [1] Wei, Xiuying, Yunchen Zhang, Xiangguo Zhang, Ruihao Gong, Shanghang Zhang, Qi Zhang, Fengwei Yu, and Xianglong Liu. "Outlier suppression: Pushing the limit of low-bit transformer language models." Advances in Neural Information Processing Systems 35 (2022): 17402-17414.
>
> [2] SPIKE NO MORE: Stabilizing the Pre-Training of Large Language Models (arXiv:2312.16903).

---

### Official Review · Reviewer_JTrQ · 2025-07-02

**Clarity:** 3
**Significance:** 2
**Originality:** 2
**Rating:** 4
**Confidence:** 4

**Summary:**

In this paper, the authors attempt to understand the realisation that deep layers in the network are less effective than actually expected. They confirm the finding across various models and experiments. They then identify the reasons related to the usage of the pre-layer normalisation. They then propose a rather simple modification of LayerNorm scaling and show that this enhances the LLM's performance.

The paper is generally well written and nice to read. I would like to kindly ask the authors a couple of questions:

1. What about other factors that affect deeper layer expressivity? Like Attention saturation, Batch sizes and so forth. Do you think those do not matter? Does it not matter as much as the cause found in the paper? Would a holistic solution consider all those together?

2. I find the analysis interesting, but can you isolate the effects of Mix-LN versus the other variant? Is it due to instability or poor parameter tuning? What could we say about this?

3. Do the results show improvements for formal reasoning problems as well? Can this help us in coding and/or formal reasoning tasks?

4. Can the authors please discuss and experiment with how their proposed method plays with "newer" tricks that we use today, like gradient checkpointing, flash attention or other memory optimisation and so forth? Would this amplify or negate what has been suggested?

5. Can the authors apply their method retroactively to high-performing models and see if they benefit? I find it lacking. It would be better to understand the actual gain we would get if possible.

All in all, i think this is a good paper but please help me understand the points above.

**Questions:**

Please see above

**Ethical Concerns:**

["NO or VERY MINOR ethics concerns only"]

**Final Justification:**

Rebuttal addressed my comments, and I remain positive about this paper.

**Limitations:**

OK

**Quality:**

3

**Strengths And Weaknesses:**

Please see above

---

> ### Author Rebuttal · Authors · 2025-07-31
>
> We sincerely appreciate your thoughtful feedback and positive evaluation. We address your remaining concerns below.
>
> **Q1. What about other factors that affect deeper layer expressivity? Like Attention saturation, Batch sizes and so forth. Do you think those do not matter? Does it not matter as much as the cause found in the paper? Would a holistic solution consider all those together?**
>
> **A1.** We appreciate the reviewer’s insightful observation regarding additional factors that may affect the expressivity of deeper layers. We do think other factors also matter for the expressivity of deeper layers, including but not limited to attention saturation [1], attention collapse [2,3,4], model dimension [5], initialization [6,7], etc. Our framework provide a simple explanation here, i.e., the (sub)-exponential growth of output variance with depth will cause deeper layers become an identify function for Pre-LN models. While other factors may affect the constant coefficient in the variance and gradient norm expressions, they do not change the growth rate. As the depth $L \to \infty$, these prefactors do not influence the asymptotic behavior.
>
>
> A holistic solution to training deep models would ideally integrate multiple factors. Our current work represents a step toward this broader goal by providing a principled intervention that is orthogonal to existing approaches and potentially complementary to them. These additional aspects are part of our ongoing research efforts and will be systematically explored in future publications.
>
>
> **Q2. I find the analysis interesting, but can you isolate the effects of Mix-LN versus the other variant? Is it due to instability or poor parameter tuning? What could we say about this?**
>
> **A2.** The divergence of Mix-LN stems from its reliance on Post-LN, which is notoriously unstable in large-scale Transformer training due to gradient vanishing issues [8,9]. While applying Post-LN in the early layers can enhance the representations of middle layers for small models compared to Pre-LN, Mix-LN becomes increasingly unstable with larger model and dataset sizes. For example, when increasing the number of training tokens from 5B (Mix-LN paper) to 10B (our paper), training LLaMa-1B diverges, as shown in Table 1.
>
>
> **Q3. Do the results show improvements for formal reasoning problems as well? Can this help us in coding and/or formal reasoning tasks?**
>
> **A3.** Thank you for your insightful question. Due to our limited computational resources, we pre-trained our models on a relatively small number of tokens, which is insufficient to achieve competitive performance on formal reasoning tasks such as code generation or mathematical problem solving.
>
> Nevertheless, it is well-established that stronger base models tend to exhibit better reasoning capabilities. Moreover, model depth plays a crucial role in reasoning, as highlighted by the Physics of Language Models paper [10]. Our proposed LNS improves the effectiveness of deeper layers, thereby enhancing the model’s capacity for complex reasoning. We are confident that our approach can lead to improved performance on formal reasoning benchmarks when applied to larger-scale models.
>
>
> **Q4. Can the authors please discuss and experiment with how their proposed method plays with "newer" tricks that we use today, like gradient checkpointing, flash attention or other memory optimisation and so forth? Would this amplify or negate what has been suggested?**
>
> **A4.**  Our approach introduces only a simple scaling operation after LayerNorm, making it, in principle, fully compatible with existing memory optimization techniques such as gradient checkpointing, mixed-precision training, and activation offloading. Thanks to its simplicity, the resulting model remains functionally equivalent to one trained with the standard Pre-LN setup, it can be integrated seamlessly into existing training pipelines without introducing additional memory overhead or architectural modifications.
>
>
> **Q5. Can the authors apply their method retroactively to high-performing models and see if they benefit? I find it lacking. It would be better to understand the actual gain we would get if possible.**
>
> **A5.** Unfortunately, since high-performing models such as Qwen and LLaMA have been trained with Pre-LN on massive datasets, it is not straightforward to apply LNS in a post-training manner. To evaluate whether the performance gains from LNS extend to more challenging and state-of-the-art LLM training settings, we conduct additional scaling experiments using OLMO, a modern open-source framework.
>
> We train models with 60M, 150M, 300M, 1B, and 7B parameters, all on a fixed 20B-token budget. As shown in the table below, LNS consistently and substantially outperforms the standard Pre-LN baseline across all model sizes. Notably, for the 7B model, LNS reduces the final training loss from 2.69 to 2.50. These results underscore the scalability of LNS and its effectiveness in large-scale pre-training regimes, highlighting its potential in large-scale pre-training scenarios.
>
>
> |  Model Size        | # Tokens | Pre-LN  | LNS  |
> |---------------|----------|--------------|-----------|
> | 60M      | 20B       |   3.67    | **3.56** |
> | 150M     | 20B       |    3.44   | **3.24** |
> | 300M     | 20B       |    3.29   | **3.14** |
> | 1B       | 20B       |   3.15    | **2.85** |
> | 7B       | 20B       |   2.69    | **2.50** |
>
> We further evaluate the generalizability of LNS by applying it to a state-of-the-art architecture, Qwen2.5-0.5B. The following table illustrates that LNS yields a notable reduction in perplexity—from 20.62 to 19.57—highlighting its effectiveness even on strong, modern architectures.
>
> |   Model      | # Tokens | Pre-LN (PPL) | LNS (PPL) |
> |---------------|----------|--------------|-----------|
> | Qwen2.5-0.5B  | 6B       | 20.62        | **19.57** |
>
> [1] Xiao, G., Tian, Y., Chen, B., Han, S. and Lewis, M., 2023. Efficient streaming language models with attention sinks. arXiv preprint arXiv:2309.17453.
>
> [2] Dong, Y., Cordonnier, J.B. and Loukas, A., 2021, July. Attention is not all you need: Pure attention loses rank doubly exponentially with depth. In International conference on machine learning (pp. 2793-2803). PMLR.
>
> [3] Geshkovski, B., Letrouit, C., Polyanskiy, Y. and Rigollet, P., 2025. A mathematical perspective on transformers. Bulletin of the American Mathematical Society, 62(3), pp.427-479.
>
> [4] Noci, Lorenzo, Sotiris Anagnostidis, Luca Biggio, Antonio Orvieto, Sidak Pal Singh, and Aurelien Lucchi. "Signal propagation in transformers: Theoretical perspectives and the role of rank collapse." Advances in Neural Information Processing Systems 35 (2022): 27198-27211.
>
> [5] Bao, Z., Yu, P., Jiang, H. and Li, Q., 2025. The Effect of Depth on the Expressivity of Deep Linear State-Space Models. arXiv preprint arXiv:2506.19296.
>
> [6] Yang, G. and Schoenholz, S., 2017. Mean field residual networks: On the edge of chaos. Advances in neural information processing systems, 30.
>
> [7] Takase, S., Kiyono, S., Kobayashi, S. and Suzuki, J., 2023. Spike no more: Stabilizing the pre-training of large language models. arXiv preprint arXiv:2312.16903.
>
> [8] Wang, H., Ma, S., Dong, L., Huang, S., Zhang, D. and Wei, F., 2024. Deepnet: Scaling transformers to 1,000 layers. IEEE Transactions on Pattern Analysis and Machine Intelligence, 46(10), pp.6761-6774.
>
> [9] Takase, S., Kiyono, S., Kobayashi, S. and Suzuki, J., 2022. B2t connection: Serving stability and performance in deep transformers. arXiv preprint arXiv:2206.00330.
>
> [10] Ye, T., Xu, Z., Li, Y. and Allen-Zhu, Z., 2024. Physics of language models: Part 2.1, grade-school math and the hidden reasoning process. arXiv preprint arXiv:2407.20311.

---

### Author Response · Authors · 2025-08-06
**Author Response**

Dear Reviewers,

We sincerely thank all reviewers for their constructive feedback and positive assessments. If there are any remaining questions or points of clarification that would be helpful for the final evaluation, we would be happy to address them. We appreciate your time and thoughtful reviews!

Best regards,

Authors

---

### Decision · Program_Chairs · 2025-09-17

**Decision:**

Accept (poster)

**Comment:**

The authors on this work start from a hypothesis that deep layers in LLM are not that effective and have limited contribution to their accuracy. They call this phenomenon 'the curse of depth'. They investigate why this is the case, and attribute this to the much used pre-layer normalization (e.g., putting the normalization before the attention as it is done in pretty much all modern LLMs). Then, they propose a solution/mitigation to this problem by scaling the contribution of layer l by 1/sqrt(l). The authors validate this both theoretically and empirically in some 'small' LLMs (130M–1B parameters).

The reviewers initially gave positive scores to the paper, giving two Borderline Accepts and 2 Accepts. In particular, most reviewers appreciate the paper's simplicity, its mathematical rigorousness and the strong experiments. However, they also find some weaknesses/unclarities that ask the authors to address. In particular:

- Reviewer JTrQ asks authors to do some analysis if this is related to other phenomenon that might make Transformers less efficient such as holistically exploring it with attention saturation and batch sizes, while asking for some other clarifications. The authors positively address these issues. Furthermore, the same reviewer asks for experiments in larger LLMs, and the authors are able to provide some preliminary results on a 7B model. The reviewer finds the arguments convincing and keeps their score.

- Reviewer skvN find some unclarities on Lemma 1 and Theorem 1, to which the authors provide a detailed explanation, that the reviewer finds convincing.

- Reviewer tprK also asks for experiments in larger networks in addition to some other experiments. The authors provide such experiments and the reviewer keeps their score.

- Reviewer L6Vr asks for some clarification on the relation of Pre-LN and LN and asks for some other experiments, that the authors happily provided, and makes the reviewer keep their score.

In general, all the reviewers remain positive about the paper and recommend it to be accepted. After reading the reviews and the paper, I agree with the reviewers and find the paper very interesting and with good results. The only thing missing is a stronger experimental setup in larger LLMs (7B+ parameters), but I acknowledge that not all labs are able to do such experiments. Nevertheless, I think that even without it, the paper easily crosses the threshold of acceptance, and the authors are encouraged to integrate the rebuttal in their camera-ready version of the paper. Congratulations to the authors!